# UM–ProtoShare: UNet-Guided, Multi-scale Shared Prototypes for Interpretable Brain Tumour Classification Using Multi-sequence 3D MRI

**Ali Golbaf**[*1] (iD)                                    ALI.GOLBAF@PLYMOUTH.AC.UK
**Vivek Singh**[1] (iD)                                    VIVEK.SINGH@PLYMOUTH.AC.UK
**Swen Gaudl**[1] (iD)                                    SWEN.GAUDL@PLYMOUTH.AC.UK
**Emmanuel Ifeachor**[1] (iD)                              E.IFEACHOR@PLYMOUTH.AC.UK

[1] *School of Engineering, Computing and Mathematics, University of Plymouth, Plymouth, UK.*

**Editors:** Accepted for publication at MIDL 2026

## Abstract

Deep learning shows strong promise in brain tumour classification using Magnetic Resonance Imaging (MRI), although limited interpretability constrains clinical translation. Most interpretability methods are post-hoc and yield visual attribution maps that are only weakly connected to the decision process. Clinicians prefer decisions built from evidence they can recognise and verify on MRI, rather than post-hoc explanations. Case-based models embed reasoning by comparing image evidence with learned prototypes, yielding "this looks like that" rationales at decision time and mirroring clinical reasoning. Building on this paradigm, we introduce UM–ProtoShare, which compares the input multi-sequence 3D brain MRI with a bank of shared, class-agnostic, multi-scale prototypes for pre-operative glioma grading. It returns not only a label, but a set of prototype matches that highlight where the model found support for its prediction. UM–ProtoShare uses a 3D ResNet-152 encoder, a lightweight UNet–style decoder with gated encoder–decoder fusions, and a normalised soft-masked mapping module to align and highlight prototype evidence on MRI. On BraTS-2020, ablations show additive benefits from the normalised mapping module, prototype sharing, multi-scale prototypes, and the decoder with gated fusions. Varying the allocation of prototypes across scales identifies a balanced accuracy–interpretability configuration that closely approaches a strong 3D ResNet-152 in classification performance (Balanced Accuracy: $88.40 \pm 2.80$; 1.48 percentage points lower) while delivering more faithful and spatially precise evidence than prior case-based models, with Activation Precision (AP) $88.72 \pm 1.60$ (+11.0% vs MProtoNet; +4.0% vs MAProtoNet) and Incremental Deletion Score (IDS) $5.10 \pm 1.30$ (lower is better, $-32.3\%$ vs MProtoNet, $-25.3\%$ vs MAProtoNet).

**Keywords:** Brain Tumour Classification, Multi-sequence 3D MRI, Interpretable Deep Learning, Case-based Models.

## 1. Introduction

Gliomas are among the most common malignant primary brain tumours (Lapointe et al., 2018). They are classified into grades I–IV based on the World Health Organization (WHO) guidelines, with higher grades reflecting increased malignancy (Louis et al., 2021). Multi-sequence Magnetic Resonance Imaging (MRI), due to its high soft-tissue contrast, is the

---

[*] Corresponding author

primary imaging modality for diagnosis, treatment planning, and long-term monitoring (Qiu et al., 2025; Sabeghi et al., 2024). Despite enhancing diagnostic reliability, the high dimensionality and heterogeneity of multi-sequence MRI, together with overlapping radiological features, make visual assessment challenging even for experienced clinicians (Qiu et al., 2025). Deep learning (DL) models, particularly Convolutional Neural Networks (CNNs), have shown promise in brain tumour assessment by extracting and analysing high-dimensional quantitative imaging features from multi-sequence brain MRI (Thomasian et al., 2022; Banerjee et al., 2019). However, most CNNs function as "black-box" models, providing limited insight into their reasoning process (Ibrahim et al., 2021). In clinical practice, where decisions directly affect patient outcomes, this opacity remains a major barrier to the adoption of DL models (Gupta et al., 2025; Rudin, 2019). Interpretability is therefore essential for the clinical translation of DL in brain tumour assessment.

Interpretability in medical imaging lacks a standardised definition (Borys et al., 2023). In line with prior work, we define interpretability as the degree to which the image structures used by a DL model align with recognisable clinical evidence. This alignment with MRI sequences and clinically meaningful concepts enables clinicians to verify, contest, and act on the model's reasoning (Rudin, 2019; Chen et al., 2022). Interpretability methods are commonly categorised into three main categories (Lu et al., 2025). (1) Visual methods generate post-hoc saliency maps but do not embed interpretability into the decision process (Rudin, 2019). They may highlight irrelevant regions resulting in misleading interpretation (Adebayo et al., 2018; Hooker et al., 2019); examples include Class Activation Mapping (CAM) (Yang et al., 2021), Gradient-based Class Activation Mapping (Grad-CAM) (Hussain and Shouno, 2023), and Layer-wise Relevance Propagation (Mandloi et al., 2024). (2) Concept-based methods map predictions to predefined clinical concepts to inform the final decision (Koh et al., 2020). However, they require increased annotation effort and are particularly demanding in medical imaging (Borys et al., 2023). (3) Case-based methods embed interpretability directly into the reasoning process by comparing image patches with learned prototypes, providing "this-looks-like-that" rationales at decision time (Chen et al., 2019). This embedded interpretability makes case-based methods particularly appealing in clinical practice (Rudin, 2019).

ProtoPNet (Chen et al., 2019) introduced case-based reasoning by learning class-specific prototypes in latent space and classifying images on the basis of patch-level similarity to these prototypes. Subsequent case-based work has improved discrimination and localisation performance, primarily in natural-image classification (Rymarczyk et al., 2021; Donnelly et al., 2022; Rymarczyk et al., 2022). However, clinical adoption of case-based methods remains limited. IAIA-BL (Barnett et al., 2021) and XProtoNet (Kim et al., 2021) integrate case-based reasoning into 2D mammography and 2D chest X-rays, respectively. Building on XProtoNet and targeting multi-sequence 3D MRI, MProtoNet (Wei et al., 2024) adds a localisation layer with soft masking and an Online-CAM loss to sharpen attention using only image-level labels. MAProtoNet (Li et al., 2024) further incorporates multi-scale feature fusion, 3D quadruplet attention, and a multi-scale mapping loss.

Despite these advances, important gaps remain for case-based models in multi-sequence 3D MRI. (1) Prototypes are class-specific, restricting reuse of MRI features shared across tumour types and grades (e.g., peritumoural oedema, necrotic cores, enhancing rims) (Louis et al., 2021; Sabeghi et al., 2024). Moreover, because the training losses push prototypes

of different classes apart in latent space, these losses can also separate semantically similar prototypes, leading to unstable predictions (Rymarczyk et al., 2021). (2) Localisation is shaped indirectly through mapping or attention modules, rather than being constrained by a spatial decoder module trained under weak supervision. (3) The effects of short- and long-range spatial relationships are largely missing in existing approaches, which either lack explicit multi-scale prototypes (derived from multi-scale feature maps) or confine multi-scale information to attention mechanisms (Li et al., 2024). This is despite the fact that tumour image features are inherently multi-scale and prone to omission without explicit multi-scale modelling (Kamnitsas et al., 2017).

To address these limitations, we introduce UM–ProtoShare, an interpretable case-based model for brain tumour assessment using multi-sequence 3D MRI. **Our contributions are:**

- **Shared and class-agnostic prototypes.** We design training objectives that learn a bank of shared, class-agnostic prototypes, where each prototype can support multiple classes via soft class–prototype coefficients derived from Grad-CAM–style importance weights. This enables efficient reuse of MRI features shared across tumour types and grades.

- **Weakly supervised localisation with gated fusions.** We improve localisation over prior case-based models by incorporating a lightweight 3D UNet–style decoder with encoder-decoder gated fusions. The proposed feature extractor produces spatially coherent features that align prototype evidence with tumour-related regions when trained using only image-level labels.

- **Explicit multi-scale prototypes.** We learn separate prototype sets for each scale of feature maps, capturing tumour appearance from fine to coarse scales and modelling short- and long-range spatial relationships. Ablations over different per-scale prototype allocations characterise how emphasising different spatial scales trades off classification accuracy against interpretability.

We position UM–ProtoShare against prior case-based models in multi-sequence 3D MRI (e.g., MProtoNet, MAProtoNet) both conceptually and architecturally, by learning a shared class-agnostic prototype bank via soft class–prototype coefficients and incorporating explicit multi-scale prototypes together with a lightweight decoder and gated encoder-decoder fusion to improve tumour-focused evidence maps across spatial scales. UM–ProtoShare embeds interpretability into the decision process, providing transparent "this looks like that" explanations while maintaining competitive classification performance and localisation coherence. The source code will be available on GitHub.

## 2. Methodology

### 2.1. Model overview

Figure 1 presents an overview of the proposed UM–ProtoShare. The UM–ProtoShare architecture comprises four main components: (1) A feature extraction backbone with a 3D encoder and a lightweight UNet–style decoder, where a gating mechanism fuses encoder and decoder features at multiple spatial scales, enhancing the focus on tumour-related regions

in the fused features. (2) A localisation component, comprising an add-on module and a mapping module. It transforms the fused features into prototype-specific features, associating each prototype with image regions that are likely to contain tumour tissue. (3) A bank of shared, class-agnostic, multi-scale prototypes that serve as reference tumour features in the latent space. Similarities between these prototypes and the prototype-specific features provide case-based image evidence at different spatial scales. (4) Lastly, a classification component aggregates prototype similarities into tumour grade predictions. The "Prototype Sharing Loss" and "Online-CAM" blocks in Figure 1 are training-only objectives and are not used during inference. The former encourages prototypes to be shared across classes, while the latter encourages tumour-focused attention maps.

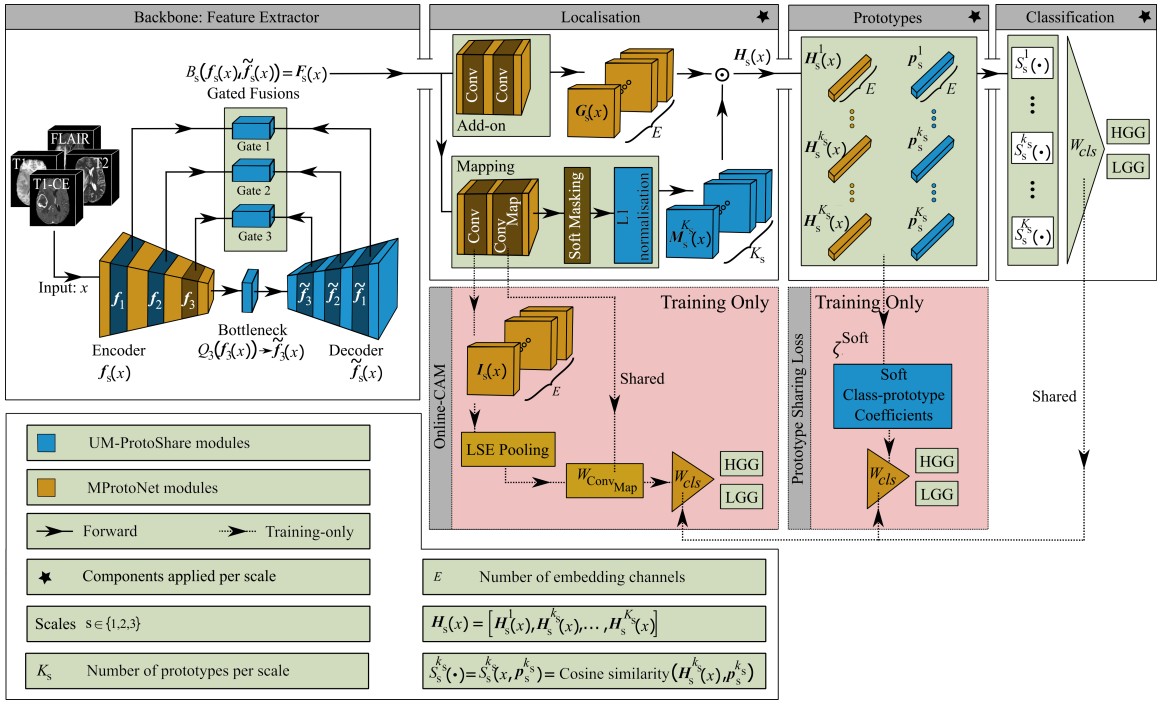

Figure 1: UM–ProtoShare design. The "Prototype Sharing Loss" and "Online-CAM" are only used during training.

## 2.2. Model architecture

**Roadmap.** For each patient, UM–ProtoShare extracts multi-sequence 3D MRI features at three spatial scales $s \in \{1, 2, 3\}$ and compares them with a bank of learned prototypes to generate its prediction. Prototypes are shared, class-agnostic, and defined at the corresponding scales $\boldsymbol{P}_s = \{\boldsymbol{p}_s^{k_s} \mid k_s \in \{1, \dots, K_s\}\}$. Given an input multi-sequence 3D MRI $x$, the backbone uses a 3D encoder to produce multi-scale features $f_s(x)$. At the deepest scale $(s = 3)$, a bottleneck transforms the encoder output into the initial decoder feature $\tilde{f}_3(x)$. The decoder then provides $\tilde{f}_s(x)$ at all scales. At each scale, a learn-

able gated fusion combines encoder and decoder features to obtain fused representations $F_s(x) = B_s(f_s(x), \tilde{f}_s(x))$. Subsequently, $F_s(x)$ is processed by the localisation component, comprising an add-on module that outputs high-level features $G_s(x)$ and a mapping module that generates per-prototype attention maps $M_s^{k_s}(x)$. Prototype-specific feature descriptors $H_s^{k_s}(x)$ are then obtained by element-wise product between $G_s(x)$ and $M_s^{k_s}(x)$ (followed by global average pooling), and compared with the corresponding prototypes in $\boldsymbol{P}_s$. Finally, cosine similarities $S_s^{k_s}(x, p_s^{k_s}) = \cos(H_s^{k_s}(x), p_s^{k_s})$ are computed and aggregated by the classifier to produce the final prediction.

**Backbone.** Given an input multi-sequence 3D brain MRI volume $x$, the 3D encoder produces feature maps $\boldsymbol{f}_s(x) \in \mathbb{R}^{C_s \times H_s \times W_s \times D_s}$ at three spatial scales $s \in \{1, 2, 3\}$, where $C_s$, $H_s$, $W_s$ and $D_s$ are the number of channels, height, width, and depth at scale $s$. At the deepest scale ($s = 3$), a bottleneck block transforms the encoder output into the initial decoder feature $\tilde{\boldsymbol{f}}_3(x) = Q_3(\boldsymbol{f}_3(x))$. Subsequent decoder blocks then produce decoder features $\tilde{\boldsymbol{f}}_s(x) \in \mathbb{R}^{C_s \times H_s \times W_s \times D_s}$ at the remaining scales $s \in \{1, 2\}$. At each scale, a gated fusion block combines encoder and decoder features to obtain fused features $\boldsymbol{F}_s(x) = B_s(\boldsymbol{f}_s(x), \tilde{\boldsymbol{f}}_s(x))$, where channel-wise gates control the relative contributions of encoder and decoder features. The fused features $\boldsymbol{F}_s(x) \in \mathbb{R}^{C_s \times H_s \times W_s \times D_s}$ are passed to the localisation component (Appendix A.1).

**Localisation component.** The localisation component consists of two modules, each of which receives the fused features $\boldsymbol{F}_s(x)$ as input. An add-on module applies a convolutional block to project $\boldsymbol{F}_s(x)$ to a fixed embedding dimension $E$, yielding high-level features $\boldsymbol{G}_s(x) \in \mathbb{R}^{E \times H_s \times W_s \times D_s}$ while preserving the spatial dimensions. In parallel, a mapping module applies a convolutional block to predict raw per-prototype attention maps $\boldsymbol{M}_s^0(x) \in \mathbb{R}^{K_s \times H_s \times W_s \times D_s}$, where $K_s$ denotes the number of prototypes at scale $s$. These maps are sharpened by a differentiable soft mask and then $\ell_1$-normalised over the spatial dimensions, yielding normalised attention maps $\boldsymbol{M}_s(x) \in \mathbb{R}^{K_s \times H_s \times W_s \times D_s}$. For each prototype $\boldsymbol{p}_s^{k_s} \in \mathbb{R}^{E \times 1 \times 1 \times 1}$, the corresponding attention map $\boldsymbol{M}_s^{k_s}(x)$ is broadcast across the $E$ channels of $\boldsymbol{G}_s(x)$ and combined via an element-wise product to produce a weighted feature map. Global average pooling over $(H_s, W_s, D_s)$ then yields a prototype-specific feature descriptor $\boldsymbol{H}_s^{k_s}(x) \in \mathbb{R}^{E \times 1 \times 1 \times 1}$ (Appendix A.2).

**Prototypes.** At each scale $s \in \{1, 2, 3\}$, UM–ProtoShare uses a set of shared, class-agnostic prototypes $\boldsymbol{P} = \bigcup_{s=1}^3 \boldsymbol{P}_s$, $\boldsymbol{P}_s = \{\boldsymbol{p}_s^{k_s} \mid k_s \in \{1, \ldots, K_s\}\}$. Each prototype $\boldsymbol{p}_s^{k_s}$ is a learnable vector in the same embedding space as $\boldsymbol{H}_s^{k_s}(x)$. Prototypes are initialised randomly and updated jointly with the other UM–ProtoShare components during training under the objectives described in Section 2.3 and Appendix B. For each prototype $\boldsymbol{p}_s^{k_s} \in \boldsymbol{P}_s$, UM–ProtoShare computes a cosine similarity $S_s^{k_s}(x, \boldsymbol{p}_s^{k_s}) = \cos(\boldsymbol{H}_s^{k_s}(x), \boldsymbol{p}_s^{k_s})$. These similarities are concatenated across all scales and prototypes into a single vector $S(x, \boldsymbol{P}) \in \mathbb{R}^{K_{\text{total}}}$, where $K_{\text{total}} = \sum_{s=1}^3 K_s$ (in our implementation, $K_{\text{total}} = 30$; see Appendix A.3).

**Classification.** The classification component is a linear layer that maps the similarity vector $S(x, \boldsymbol{P})$ to class logits. Let $C$ denote the number of classes and $\boldsymbol{w}_{\text{cls}} \in \mathbb{R}^{C \times K_{\text{total}}}$ the classifier weights. For class $c$, the logit $y_c(x)$ is obtained by a weighted sum of prototype similarities $y_c(x) = \sum_{s=1}^3 \sum_{k_s=1}^{K_s} w_{\text{cls}}[c, s, k_s] S_s^{k_s}(x, \boldsymbol{p}_s^{k_s})$, where $w_{\text{cls}}[c, s, k_s]$ denotes

the classifier weight associated with prototype $\boldsymbol{p}_s^{k_s}$ for class $c$. A softmax over $\{y_c(x)\}_{c=1}^C$ yields the predicted class probabilities (Appendix A.4).

## 2.3. Training objectives and loss functions

UM–ProtoShare follows the training objectives used in case-based models (Chen et al., 2019), adapted to shared, multi-scale prototypes. The goal is to learn a latent space in which prototype-specific descriptors $\boldsymbol{H}_s^{k_s}(x)$ capture tumour-related imaging features and their associated regions, to anchor each prototype $\boldsymbol{p}_s^{k_s}$ to a specific 3D region of a real training image, and to train a classifier that maps the full similarity vector $S(x, \boldsymbol{P})$ to class logits.

To realise these objectives, UM–ProtoShare adopts the three-stage training procedure used in prior case-based models (Chen et al., 2019): (1) latent space learning; (2) prototype reassignment ("push"); and (3) classifier training. Stages (1) and (3) update model parameters via gradient-based optimisation, while Stage (2) performs prototype reassignment on the training set.

**Stage (1): Latent space learning.** In Stage (1), we optimise all components preceding the classification layer by minimising:

$$L_{\text{stage 1}} = \lambda_{\text{cls}} L_{\text{cls}} + \lambda_{\text{clst}} L_{\text{clst}}^{\text{class-agnostic}} + \lambda_{\text{sep}} L_{\text{sep}}^{\text{class-agnostic}} + \frac{\lambda_{\text{map}}}{3} \sum_{s=1}^{3} L_{\text{map}}^s + \frac{\lambda_{\text{OC}}}{3} \sum_{s=1}^{3} L_{\text{OC}}^s + \lambda_{\text{div}} L_{\text{div}}. \tag{1}$$

Where, $L_{\text{cls}}$ is the classification loss. $L_{\text{clst}}^{\text{class-agnostic}}$ and $L_{\text{sep}}^{\text{class-agnostic}}$ are the class-agnostic clustering and separation losses, reformulated from the standard class-specific ProtoPNet formulation (Chen et al., 2019) to be compatible with shared prototypes while preserving class-discriminative learning ($L_{\text{clst}}^{\text{class-specific}} \rightarrow L_{\text{clst}}^{\text{class-agnostic}}$ and $L_{\text{sep}}^{\text{class-specific}} \rightarrow L_{\text{sep}}^{\text{class-agnostic}}$). The mapping and Online-CAM losses are adapted to the multi-scale setting, yielding scale-specific terms $L_{\text{map}}^s$ and $L_{\text{OC}}^s$, following MProtoNet (Wei et al., 2024). We also introduce a prototype-diversity regulariser $L_{\text{div}}$ to reduce redundancy in the prototype bank by discouraging highly similar prototypes at each scale. The coefficients $\lambda.$ are scalar loss weights that balance the different training objectives; all values are fixed across folds and experiments and are reported in Appendix D.

**Stage (2): Prototype reassignment ("push").** In Stage (2), each prototype is reassigned to the closest matching prototype-specific descriptor extracted from the training set:

$$\boldsymbol{p}_s^{k_s} \leftarrow \boldsymbol{H}_s^{k_s}(x_{n'}), \quad n' = \arg \max_{1 \leq n \leq N} S_s^{k_s}(x_n, \boldsymbol{p}_s^{k_s}). \tag{2}$$

Here, $\{x_n\}_{n=1}^N$ denotes the training set, and $n'$ indexes the training sample whose descriptor maximises the similarity to prototype $\boldsymbol{p}_s^{k_s}$ at scale $s$.

**Stage (3): Classifier training.** In Stage (3), we optimise the classification component while keeping the remaining network fixed by minimising:

$$L_{\text{stage 3}} = \lambda_{\text{cls}} L_{\text{cls}} + \lambda_{L_1} L_{L_1}. \tag{3}$$

Here, $L_{L_1}$ is an $\ell_1$-regularisation loss on the classifier weights, following ProtoPNet (Chen et al., 2019). Further details of the training stages are provided in Appendix B.1.

**Clustering and separation losses.** As described in Section 2.2, UM–ProtoShare maps each input $x$ to prototype-specific descriptors $\boldsymbol{H}_s^{k_s}(x)$ and computes their cosine similarities to the corresponding prototypes, $S_s^{k_s}(x, \boldsymbol{p}_s^{k_s}) = \cos(\boldsymbol{H}_s^{k_s}(x), \boldsymbol{p}_s^{k_s})$. The clustering and separation losses encourage high similarity between descriptors and prototypes that support the ground-truth class, and low similarity to prototypes that support other classes. The formulation of these losses determines whether prototypes are class-specific or class-agnostic.

In standard ProtoPNet–style training, prototypes are class-specific and rely on hard class–prototype ownership coefficients $\zeta^{\mathrm{hard}} \in \{0, 1\}$ to define the clustering and separation losses (Chen et al., 2019) (see Appendix B.2). UM–ProtoShare uses shared, class-agnostic prototypes, in which each prototype can support multiple classes. This reflects the fact that several MRI features (e.g., peritumoural oedema, necrotic cores, enhancing rims) may legitimately appear across tumour types and grades (Louis et al., 2021; Sabeghi et al., 2024). This choice affects only the training objectives. At inference, the classifier always uses the full similarity vector $S(x, \boldsymbol{P})$ to deliver the final prediction.

Shared, class-agnostic prototypes that retain class-discriminative power can be understood by analogy to class activation mapping methods such as Grad-CAM. These methods operate on a shared set of feature maps in the latent space and, for a chosen class $c$, compute class-discriminative importance weights $\alpha_{c,k}(x)$ for each feature map to form class-discriminative localisation maps. In UM–ProtoShare, we similarly introduce class-discriminative importance weights $\gamma_{c,\boldsymbol{p}_s^{k_s}}$ for each shared prototype $\boldsymbol{p}_s^{k_s}$, in direct analogy to the Grad-CAM weights for feature maps. Our formulation shows that these importance weights coincide with the classifier weights and thus quantify how much each prototype contributes to the class logit $y_c(x)$. From these importance weights, we derive soft class–prototype coefficients $\zeta_{c,\boldsymbol{p}_s^{k_s}}^{\mathrm{soft}}$, which encode how strongly each prototype supports each class and serve as soft ownership factors in the class-agnostic clustering and separation losses.

In Grad-CAM, given feature maps $\boldsymbol{Z}^k(x) \in \mathbb{R}^{H \times W \times D}$, $k \in \{1, \ldots, K\}$, and a class score $y_c(x)$, the importance weight for each feature map is defined as (Selvaraju et al., 2017),

$$\alpha_{c,k}(x) = \frac{1}{HWD} \sum_{m=1}^{H} \sum_{n=1}^{W} \sum_{l=1}^{D} \frac{\partial y_c(x)}{\partial Z_{m,n,l}^k(x)}. \tag{4}$$

Global average pooling of the gradients over the spatial dimensions yields one scalar importance weight per feature map.

In UM–ProtoShare, the classification component operates on prototype similarities rather than feature maps,

$$y_c(x) = \sum_{s=1}^{3} \sum_{k_s=1}^{K_s} w_{\mathrm{cls}}[c, s, k_s] \, S_s^{k_s}(x, \boldsymbol{p}_s^{k_s}). \tag{5}$$

The term $w_{\mathrm{cls}}[c, s, k_s]$ denotes the classifier weight linking prototype $\boldsymbol{p}_s^{k_s}$ to class $c$. In direct analogy to the Grad-CAM importance weights in Equation (4), we define the class-discriminative importance weight of prototype $\boldsymbol{p}_s^{k_s}$ for class $c$ as the derivative of the class score with respect to its similarity,

$$\gamma_{c,\boldsymbol{p}_s^{k_s}} = \frac{\partial y_c(x)}{\partial S_s^{k_s}(x, \boldsymbol{p}_s^{k_s})} = w_{\mathrm{cls}}[c, s, k_s]. \tag{6}$$

Here, prototype similarities $S_s^{k_s}(x, \boldsymbol{p}_s^{k_s})$ are already spatially pooled scalar quantities derived from feature descriptors (see Equation (14) and Equation (15) in Appendix A), so the Grad-CAM–style definition of importance weights in Equation (4) reduces to taking the derivative of the class score with respect to $S_s^{k_s}(x, \boldsymbol{p}_s^{k_s})$, as in Equation (6). Moreover, because the classifier is linear in the prototype similarities (see Equation (5)), these importance weights coincide with the classifier weights.

The importance weights $\gamma_{c,\boldsymbol{p}_s^{k_s}}$ thus play the same role as the Grad-CAM weights $\alpha_{c,k}(x)$, but at the level of shared prototypes rather than feature maps, indicating how each prototype influences the logit for class $c$. We then construct non-negative, normalised soft class–prototype coefficients from these importance weights,

$$\zeta_{c,\boldsymbol{p}_s^{k_s}}^{\text{soft}} = \frac{\text{ReLU}\big(w_{\text{cls}}[c, s, k_s]\big)}{\sum_{c'} \text{ReLU}\big(w_{\text{cls}}[c', s, k_s]\big) + \varepsilon}, \quad \varepsilon = 10^{-6}, \ \text{ReLU}(x) = \max(x, 0). \tag{7}$$

By construction, $\zeta_{c,\boldsymbol{p}_s^{k_s}}^{\text{soft}} \in [0, 1]$ and $\sum_c \zeta_{c,\boldsymbol{p}_s^{k_s}}^{\text{soft}} = 1$, so for each prototype $\boldsymbol{p}_s^{k_s}$ the vector $\{\zeta_{c,\boldsymbol{p}_s^{k_s}}^{\text{soft}}\}_c$ defines a soft ownership distribution over classes. The ReLU ensures that only positive classifier weights are treated as support when assigning a prototype to classes, mirroring the focus on positive evidence for the class of interest in Grad-CAM.

Using these coefficients, the class-agnostic clustering and separation losses for an input $(x, y = c)$ across scales are defined as,

$$L_{\text{clst}}^{\text{class-agnostic}} = -\sum_{s=1}^{3} \sum_{k_s=1}^{K_s} \zeta_{c,\boldsymbol{p}_s^{k_s}}^{\text{soft}} S_s^{k_s}(x, \boldsymbol{p}_s^{k_s}). \tag{8}$$

$$L_{\text{sep}}^{\text{class-agnostic}} = \sum_{s=1}^{3} \sum_{k_s=1}^{K_s} \Big(\sum_{c' \neq c} \zeta_{c',\boldsymbol{p}_s^{k_s}}^{\text{soft}}\Big) S_s^{k_s}(x, \boldsymbol{p}_s^{k_s}). \tag{9}$$

Thus, every prototype $\boldsymbol{p}_s^{k_s}$ with non-zero $\zeta_{c,\boldsymbol{p}_s^{k_s}}^{\text{soft}}$ receives a training gradient, scaled by how strongly it supports class $c$. This yields smoother and more stable optimisation than hard class–prototype assignments, particularly in the shared multi-scale setting. In the binary case, the separation weight simplifies to $\sum_{c' \neq c} \zeta_{c',\boldsymbol{p}_s^{k_s}}^{\text{soft}} = 1 - \zeta_{c,\boldsymbol{p}_s^{k_s}}^{\text{soft}}$.

**Prototype-diversity regulariser loss.** Without additional constraints, multiple prototypes in the shared bank may collapse onto similar regions of the latent space, yielding redundant explanations and thereby limiting its representational capacity. To mitigate this, we introduce a prototype-diversity regulariser $L_{\text{div}}$ that penalises the average pairwise cosine similarity between normalised prototypes at each scale,

$$L_{\text{div}} = \frac{1}{3} \sum_{s=1}^{3} \text{Mean}_{1 \leq i < j \leq K_s} \left(\frac{\boldsymbol{p}_s^i \cdot \boldsymbol{p}_s^j}{\|\boldsymbol{p}_s^i\| \, \|\boldsymbol{p}_s^j\|}\right). \tag{10}$$

Here $\boldsymbol{p}_s^i$ and $\boldsymbol{p}_s^j$ denote the $i$-th and $j$-th prototypes at scale $s$, and the mean is taken over all prototype pairs. Minimising $L_{\text{div}}$ encourages prototypes at each scale to span diverse directions in the latent space, reducing redundancy in the prototype bank and promoting coverage of distinct tumour features, in line with diversity regularisation used in recent case-based models (Rymarczyk et al., 2022; Ayoobi et al., 2025; Wei and Zhu, 2025). This cosine penalty aligns with the cosine similarity used for prototype matching.

## 3. Experiments and results

### 3.1. Dataset

We evaluated UM–ProtoShare using the BraTS-2020 training cohort (Menze et al., 2014), a multi-sequence 3D brain MRI dataset comprising 369 pre-operative glioma cases, each with four sequences, including T1-weighted (T1), contrast-enhanced T1-weighted (T1-CE), T2-weighted (T2), and T2 Fluid-Attenuated Inversion Recovery (FLAIR) MRI. WHO tumour grades were provided (low-grade gliomas (LGG): 76, high-grade gliomas (HGG): 293). Ground-truth segmentations were also available for three subregions, including enhancing tumour, peritumoural oedema, and necrotic core, which we merged into a single tumour mask. Only MRI scans and tumour grades were used for training, while tumour masks were used exclusively to compute interpretability metrics. Preprocessing and data augmentation procedures are described in Appendix C.

### 3.2. Evaluation metrics and training hyper-parameters

We evaluate the classification performance of UM–ProtoShare using the balanced accuracy (BAC) metric, which compensates for class imbalance (Wei et al., 2024). Interpretability is assessed in terms of localisation coherence and correctness, measured respectively by Activation Precision (AP; higher is better) and Incremental Deletion Score (IDS; lower is better) (Nauta et al., 2023). All results are reported using 5-fold cross-validation on different training and test splits. Full definitions of the evaluation metrics and the complete set of training hyper-parameters are provided in Appendix D.

### 3.3. Ablation studies

To systematically examine the design choices in UM–ProtoShare, we perform a series of ablations A1-A2, B, C, D and E1-E3 under a fixed total prototype budget $K_{\text{total}} = \sum_{s=1}^{3} K_s = 30$ and identical training settings (same data splits, optimiser and loss weights). The ablations progressively introduce the main components of UM–ProtoShare: A1 is a single-scale, class-specific prototype model without the decoder and gated fusion; A2 replaces class-specific prototypes by shared, class-agnostic prototypes; B introduces explicit multi-scale prototypes; C adds the decoder module; D enables gated encoder-decoder fusion; and E1-E3 vary the per-scale prototype allocation (fine-heavy, mid-heavy and coarse-heavy) while keeping $K_{\text{total}}$ fixed. For each configuration, we evaluate the effects on classification performance (BAC) and interpretability (AP, IDS). Table 1 reports the results for all ablation variants. Full architectural details for each ablation configuration are provided in Appendix E.

Starting from A1 and progressively introducing UM–ProtoShare components (A2-E), BAC remains competitive and reaches values between $85.92 \pm 2.14$ and $89.21 \pm 3.18$, with the coarse-heavy configuration (E3) achieving the highest BAC among the ablations. Importantly, the ablations reveal consistent trends that clarify the role of each component. Moving from class-specific to class-agnostic prototypes (A1→A2) improves BAC, indicating that prototype sharing across classes is beneficial in this setting where tumour features overlap across LGG/HGG. Introducing explicit multi-scale prototypes (A2→B) yields a further BAC gain, confirming that aggregating evidence across spatial scales is helpful for

glioma grading. Adding the decoder module (B→C) trades some BAC for stronger localisation behaviour, while incorporating gated fusions (C→D) recovers most of the lost accuracy while keeping the locality benefits of the decoder in place.

Across these configurations, interpretability improves markedly. AP increases from $81.38 \pm 2.23$ (A1) to $90.10 \pm 1.80$ (E1), while IDS decreases from $7.13 \pm 2.93$ to $4.60 \pm 1.20$, demonstrating progressively more tumour-focused and faithful evidence maps as components are introduced. In particular, prototype sharing (A2) provides an initial AP improvement, explicit multi-scale prototypes (B) improve both AP and IDS, and the decoder (C) produces the largest reduction in IDS, sharpening tumour-focused localisation under weak supervision. Finally, reallocating prototypes across scales (E1-E3) modulates the accuracy–interpretability balance under a fixed prototype budget. The fine-heavy allocation (E1) yields the strongest localisation, the coarse-heavy variant (E3) favours higher BAC, and the mid-heavy configuration (E2) provides the most balanced operating point. Using E2 as our default UM–ProtoShare model, we obtain $88.40 \pm 2.80$ BAC, $88.72 \pm 1.60$ AP and $5.10 \pm 1.30$ IDS.

Table 1: Classification and interpretability performance of UM–ProtoShare and baseline models on BraTS-2020 (mean $\pm$ std over five folds; higher BAC/AP and lower IDS are better).

| Model | Ablation | Design configuration | BAC (%, ↑) | IDS (%, ↓) | AP (%, ↑) |
|---|---|---|---|---|---|
| CNN: Grad-CAM | – | – | $89.73 \pm 2.36$ | $14.85 \pm 5.21$ | $10.65 \pm 5.63$ |
| CNN: Occlusion | – | – | $89.73 \pm 2.36$ | $10.43 \pm 4.59$ | $15.63 \pm 3.84$ |
| ProtoPNet | – | – | $86.42 \pm 1.78$ | $21.56 \pm 5.83$ | $11.73 \pm 3.02$ |
| XProtoNet | – | – | $87.23 \pm 1.59$ | $17.46 \pm 4.98$ | $18.12 \pm 2.76$ |
| MProtoNet | – | – | $84.69 \pm 3.03$ | $7.53 \pm 3.24$ | $79.93 \pm 1.32$ |
| MAProtoNet | – | – | $87.36 \pm 3.26$ | $6.83 \pm 3.12$ | $85.29 \pm 4.14$ |
| | A1 | single-scale; class-specific | $85.92 \pm 2.14$ | $7.13 \pm 2.93$ | $81.38 \pm 2.23$ |
| | A2 | single-scale; class-agnostic | $87.43 \pm 1.27$ | $7.49 \pm 4.07$ | $83.28 \pm 3.41$ |
| | B | multi-scale; class-agnostic | $88.85 \pm 2.56$ | $6.52 \pm 3.29$ | $85.89 \pm 1.24$ |
| UM–ProtoShare | C | multi-scale; class-agnostic; decoder | $86.47 \pm 2.85$ | $5.23 \pm 2.18$ | $87.51 \pm 1.38$ |
| | D | multi-scale; class-agnostic; decoder; gated fusions | $88.14 \pm 2.74$ | $5.36 \pm 1.28$ | $86.45 \pm 1.38$ |
| | E1 | fine-heavy | $87.90 \pm 2.90$ | $4.60 \pm 1.20$ | $90.10 \pm 1.80$ |
| | E2 | mid-heavy | $88.40 \pm 2.80$ | $5.10 \pm 1.30$ | $88.72 \pm 1.60$ |
| | E3 | coarse-heavy | $89.21 \pm 3.18$ | $5.98 \pm 2.57$ | $85.80 \pm 2.08$ |

## 3.4. Comparative benchmark

We compare UM–ProtoShare against a CNN model derived from the UM–ProtoShare backbone (evaluated with Grad-CAM and Occlusion saliency), as well as ProtoPNet, XProtoNet, MProtoNet and MAProtoNet. All models are trained under the same data splits and comparable optimisation settings. Implementation details for each baseline are provided in Appendix F. Quantitative results are summarised in Table 1, and representative prototype attention maps are shown in Figure 2.

The comparative benchmark reveals a clear accuracy–interpretability trade-off between purely discriminative and case-based reasoning. Across post-hoc saliency methods, the CNN achieves the highest BAC ($89.73 \pm 2.36$), but exhibits poor localisation behaviour under both Grad-CAM and Occlusion, with Occlusion yielding marginally better AP/IDS,

highlighting the limited explanatory value of saliency under a purely discriminative objective. In contrast, case–based models provide substantially stronger localisation, confirming that prototype-based evidence aligns better with tumour-related regions.

UM–ProtoShare achieves $88.40 \pm 2.80$ BAC, $88.72 \pm 1.60$ AP and $5.10 \pm 1.30$ IDS, providing the strongest overall accuracy–interpretability operating point among the evaluated methods. Among prior baselines, MAProtoNet is the most competitive in this setting, while older approaches such as ProtoPNet and XProtoNet show markedly weaker tumour localisation. Compared with MAProtoNet, UM–ProtoShare improves AP and reduces IDS (approximately a 25% relative reduction) while maintaining competitive BAC. Relative to MProtoNet and older case-based approaches, UM–ProtoShare yields consistent gains in interpretability (higher AP, lower IDS) while retaining competitive classification performance. Finally, compared with the backbone CNN, UM–ProtoShare trades a small BAC reduction for substantially stronger, tumour-focused evidence maps, maintaining competitive classification performance.

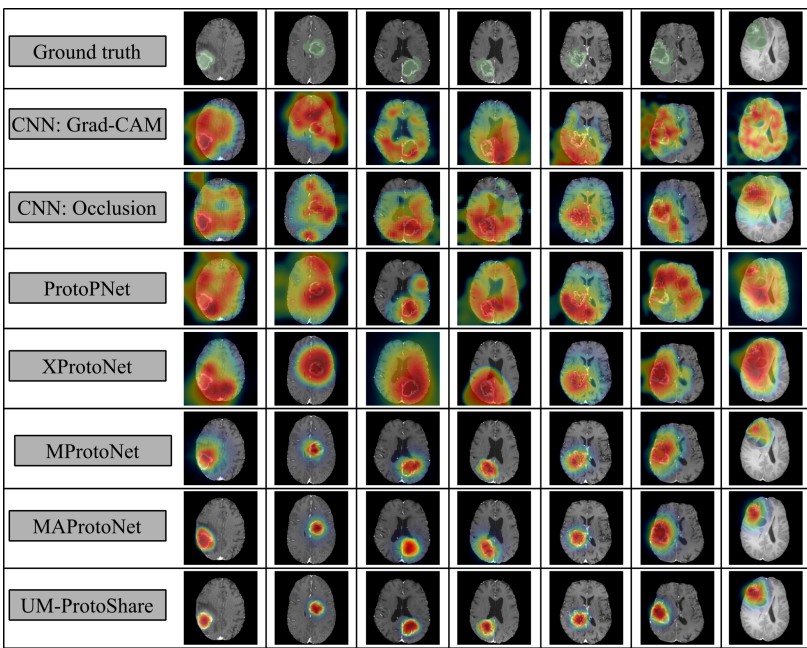

Figure 2: Attribution maps for representative BraTS-2020 cases (T1-CE slices). UM–ProtoShare provides more precise tumour localisation than prior case-based models.

## 4. Discussion

**Design rationale.** UM–ProtoShare is designed to support clinically plausible case-based reasoning, where evidence should remain tumour-focused, comparable across cases, and interpretable across scales. The normalised mapping module attenuates tumour-size effects

in attention values, improving comparability across volumes such that evidence better reflects where the model attends rather than how large the lesion is (Xu et al., 2022). The UNet–style decoder recovers spatial detail lost through encoder downsampling and supports boundary-consistent localisation, while gated encoder-decoder fusion regulates scale-wise feature contributions to promote clearer, scale-coherent evidence maps (Oktay et al., 2018; Hu et al., 2018; Takikawa et al., 2019). Class-agnostic prototype sharing enables reuse of appearance patterns that may occur across LGG/HGG in multi–sequence MRI, improving data efficiency by reducing redundancy in the prototype space (Haydar et al., 2022; Rymarczyk et al., 2021). Potential over-sharing towards majority-class patterns is mitigated by soft, class-conditioned assignment in the separation loss and the introduced diversity regulariser (Gautam et al., 2023). An extended discussion is provided in Appendix H.

**Qualitative interpretability.** Our qualitative analysis (Figure 5) highlights that prototype similarity should not be interpreted as direct class membership, since tumour-related MRI cues may overlap across LGG/HGG and remain ambiguous in isolation. Instead, UM–ProtoShare disentangles pattern reusability from diagnostic effect: shared class-agnostic prototypes can strongly match tumour-related regions across grades, while class-specific classifier weights determine whether the same matched evidence acts as supporting evidence or counter-evidence for a given prediction. This behaviour is clinically meaningful, as it reflects how radiologists rely on multi-sequence context and multi-scale evidence to disambiguate partially shared cues. An extended qualitative analysis and clinical interpretation are provided in Appendix G.

**Limitations and future work.** Although the dataset is multi-institutional, the cohort remains limited for 3D multi-sequence MRI, particularly for LGG; larger cohorts and external validation are required to assess robustness and generalisability. Prototypes are learned from stacked multi-sequence volumes, which simplifies training but can obscure which sequence drives a match and may allow occasional off-target evidence outside clinically important subregions. Future work will investigate sequence-wise prototype banks to disentangle modality- and region-specific evidence, and evaluate UM–ProtoShare on additional classification tasks (including multi-class settings) for stronger interpretability assessment. Integrating concept-based approaches could also attach higher-level clinical semantics to prototype evidence while preserving case-based reasoning.

## 5. Conclusion

UM–ProtoShare combines shared class-agnostic multi-scale prototypes with a normalised mapping module and a decoder with gated fusions to achieve a favourable accuracy–interpretability balance for glioma grading (LGG/HGG) on multi-sequence 3D MRI. On BraTS-2020, it attains competitive balanced accuracy compared with a strong CNN baseline while substantially improving quantitative interpretability metrics. Each prediction is supported by spatially localised prototype matches, yielding directly inspectable evidence maps in a "this looks like that" form. Taken together, these properties make UM–ProtoShare a practical step toward clinically legible, case-based decision support for brain tumour assessment.

## Acknowledgments

This work was supported by a UK EPSRC studentship and, in part, by Brain Tumour Research.

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

## Appendix A. Model architecture

### A.1. Backbone

**Encoder.** UM–ProtoShare uses a 3D extension of ResNet-152 (He et al., 2016) as the encoder backbone, a well-established model for brain tumour classification using MRI (Gupta et al., 2024; Kong et al., 2025). All 2D components (convolutions, batch normalisation, pooling, residual blocks) are converted to their 3D counterparts. To preserve localisation, the 3D ResNet-152 is truncated after the second residual stage (Wei et al., 2024), yielding three feature scales $s \in \{1, 2, 3\}$. These correspond to the outputs of the initial stage, residual stage (1), and residual stage (2) (see Figure 3). We denote these feature vectors by $\boldsymbol{f}_s(x) \in \mathbb{R}^{C_s \times H_s \times W_s \times D_s}$, where $C_s$, $H_s$, $W_s$ and $D_s$ are the number of channels, height, width and depth at scale $s$. Inspired by (Ebrahimi et al., 2020), we employ transfer learning by initialising the 3D encoder from ImageNet-pretrained 2D ResNet-152 weights. To this end, convolutional filters are repeated along the depth axis to form 3D kernels, and the first convolution is adapted to four-channel input by copying the pretrained weights for the first three channels and initialising the fourth channel with their mean (Cui et al., 2018).

**Decoder.** To enhance locality, we append a lightweight 3D UNet–style decoder to the encoder. As depicted in Figure 3, it comprises a bottleneck block at the deepest spatial scale to stabilise the input (Ahir and Parikh, 2025), followed by two decoder blocks that each perform trilinear upsampling, a convolution, batch normalisation and ReLU activation. No skip connections are used inside the decoder. We adopt interpolation-based upsampling rather than transposed convolution because transposed convolutions can introduce uneven overlap artefacts, whereas trilinear upsampling avoids these artefacts and is more computationally efficient in decoder stages for medical imaging analysis (Sanjar et al., 2020; Alwadee et al., 2025). The resulting decoder features are denoted by $\tilde{\boldsymbol{f}}_s(x) \in \mathbb{R}^{C_s \times H_s \times W_s \times D_s}$ for $s \in \{1, 2, 3\}$, with shapes matched to the encoder features at each scale to enable gated fusion.

**Gated fusion.** We fuse encoder features $\boldsymbol{f}_s(x)$ and decoder features $\tilde{\boldsymbol{f}}_s(x)$ using per-channel gated fusion mechanisms (Oktay et al., 2018; Yu et al., 2018). As illustrated in Figure 3, for each scale the encoder and decoder features are first concatenated, globally averaged over the spatial dimensions, and then passed through a gating block composed of two convolutional layers with a ReLU activation in between, as in (Yu et al., 2018). A sigmoid activation then yields a per-channel gate vector $\boldsymbol{\alpha}_s \in (0, 1)^{C_s}$, which is used as per-channel weights to modulate the relative contributions of the corresponding features,

$$\boldsymbol{F}_s(x) = \boldsymbol{\alpha}_s \odot \boldsymbol{f}_s(x) + \left(1 - \boldsymbol{\alpha}_s\right) \odot \tilde{\boldsymbol{f}}_s(x). \tag{11}$$

Where $\odot$ denotes channel-wise multiplication. At each scale, the fused features $\boldsymbol{F}_s(x) \in \mathbb{R}^{C_s \times H_s \times W_s \times D_s}$ preserve the shapes of $\boldsymbol{f}_s(x)$ and $\tilde{\boldsymbol{f}}_s(x)$.

### A.2. Localisation component

Following prior case-based models (Wei et al., 2024; Li et al., 2024), the localisation component comprises two modules: an add-on module and a mapping module, as presented in Figure 4.

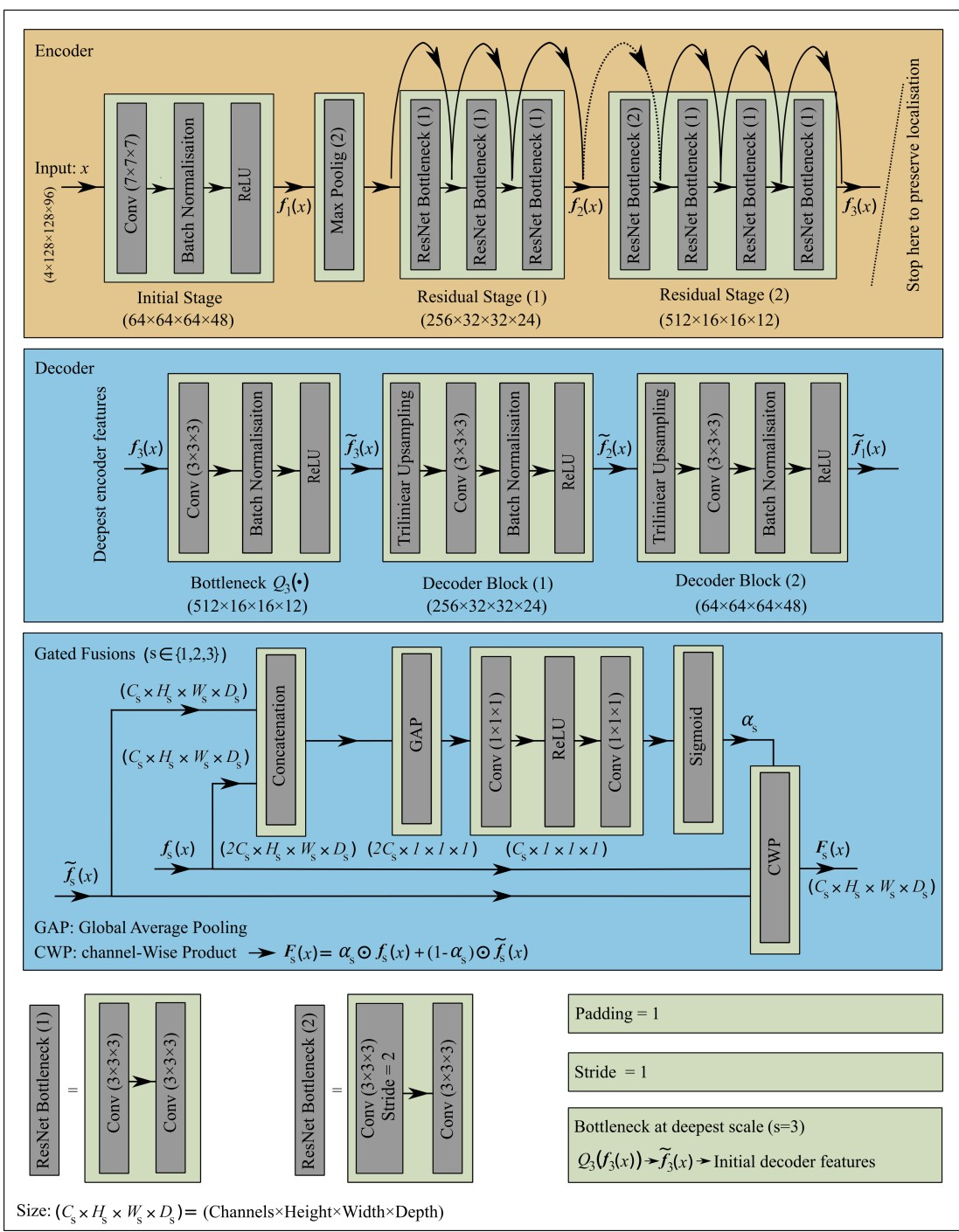

Figure 3: Backbone design. A truncated 3D ResNet-152 encoder produces three feature scales $\boldsymbol{f}_s(x) \in \mathbb{R}^{C_s \times H_s \times W_s \times D_s}$ for $s \in \{1, 2, 3\}$. A lightweight UNet–style decoder with a bottleneck and two decoder blocks produces matched decoder features $\tilde{\boldsymbol{f}}_s(x)$, and gated fusions combine $\boldsymbol{f}_s(x)$ and $\tilde{\boldsymbol{f}}_s(x)$ at each scale to produce the final feature maps $\boldsymbol{F}_s(x)$.

**Add-on module.** At each scale, a convolutional block comprising convolution operations, batch normalisation and ReLU activation projects the fused features $\boldsymbol{F}_s(x) \in \mathbb{R}^{C_s \times H_s \times W_s \times D_s}$ to a fixed embedding dimension while preserving the spatial dimensions, yielding $\boldsymbol{G}_s(x) \in \mathbb{R}^{E \times H_s \times W_s \times D_s}$. This acts as a channel-harmonising projection, with $(H_s, W_s, D_s)$ unchanged. We set $E = 128$ at all scales to ensure simplicity and comparability.

**Mapping module.** At each scale, a convolutional block consisting of convolutions, batch normalisation and ReLU activation predicts per-prototype attention maps. We denote the raw stack of these maps by $\boldsymbol{M}_s^0(x) \in \mathbb{R}^{K_s \times H_s \times W_s \times D_s}$, and the map for a single prototype $\boldsymbol{p}_s^{k_s}$ by $\boldsymbol{M}_s^{0,k_s}(x) \in \mathbb{R}^{H_s \times W_s \times D_s}$. A sigmoid activation rescales the values of $\boldsymbol{M}_s^{0,k_s}(x)$ to the range of $[0, 1]$. To sharpen attention maps and suppress irrelevant background, we further apply a differentiable soft mask (Li et al., 2018):

$$\boldsymbol{M}_s^{1,k_s}(x) = \frac{1}{1 + \exp\left(-w\left(\boldsymbol{M}_s^{0,k_s}(x) - \sigma\right)\right)}. \tag{12}$$

Where $w$ and $\sigma$ are hyperparameters, set to $w = 10$ and $\sigma = 0.5$ (Li et al., 2018). We then obtain size-invariant attention maps by $\ell_1$-normalisation over the spatial dimensions:

$$\boldsymbol{M}_s^{k_s}(x) = \frac{\boldsymbol{M}_s^{1,k_s}(x)}{\sum \boldsymbol{M}_s^{1,k_s}(x) + \varepsilon}, \quad \varepsilon = 10^{-6}. \tag{13}$$

Where the sum is taken over all spatial locations. Stacking the normalised per-prototype maps over $k_s$ yields $\boldsymbol{M}_s(x) \in \mathbb{R}^{K_s \times H_s \times W_s \times D_s}$.

**Prototype descriptors.** For each prototype $\boldsymbol{p}_s^{k_s}$ at scale $s$, the corresponding attention map $\boldsymbol{M}_s^{k_s}(x)$ is broadcast across the $E$ channels of $\boldsymbol{G}_s(x) \in \mathbb{R}^{E \times H_s \times W_s \times D_s}$. We then perform a voxel-wise (elementwise) product and apply global average pooling over the spatial dimensions $(H_s, W_s, D_s)$ to obtain the corresponding feature descriptor $\boldsymbol{H}_s^{k_s}(x) \in \mathbb{R}^{E \times 1 \times 1 \times 1}$. Stacking descriptors across prototypes yields $\boldsymbol{H}_s(x) \in \mathbb{R}^{K_s \times E \times 1 \times 1 \times 1}$, which serves as the per-scale input to the prototype component of the model. Formally,

$$\boldsymbol{H}_s^{k_s}(x) = \frac{1}{H_s W_s D_s} \sum_{m=1}^{H_s} \sum_{n=1}^{W_s} \sum_{\ell=1}^{D_s} \left(\boldsymbol{G}_s(x) \odot \boldsymbol{M}_s^{k_s}(x)\right)_{m,n,\ell}. \tag{14}$$

Where $\odot$ denotes elementwise multiplication and the subscript $(m, n, \ell)$ indexes spatial location $(m, n, \ell)$ across all $E$ channels.

### A.3. Prototypes

**Prototype bank.** UM–ProtoShare uses a bank of shared, class-agnostic, multi-scale prototypes. For scales $s \in \{1, 2, 3\}$, the bank is defined as

$$\boldsymbol{P} = \bigcup_{s=1}^{3} \boldsymbol{P}_s, \quad \boldsymbol{P}_s = \left\{\boldsymbol{p}_s^{k_s} \mid k_s \in \{1, \ldots, K_s\}\right\}. \tag{15}$$

Where each prototype $\boldsymbol{p}_s^{k_s} \in \mathbb{R}^{E \times 1 \times 1 \times 1}$ is a learnable vector. Following prior case-based models (Wei et al., 2024; Li et al., 2024), we fix the total prototype budget to $K_{\text{total}} = \sum_{s=1}^{3} K_s = 30$ to ensure a fair, capacity-matched comparison across ablations and baseline models.

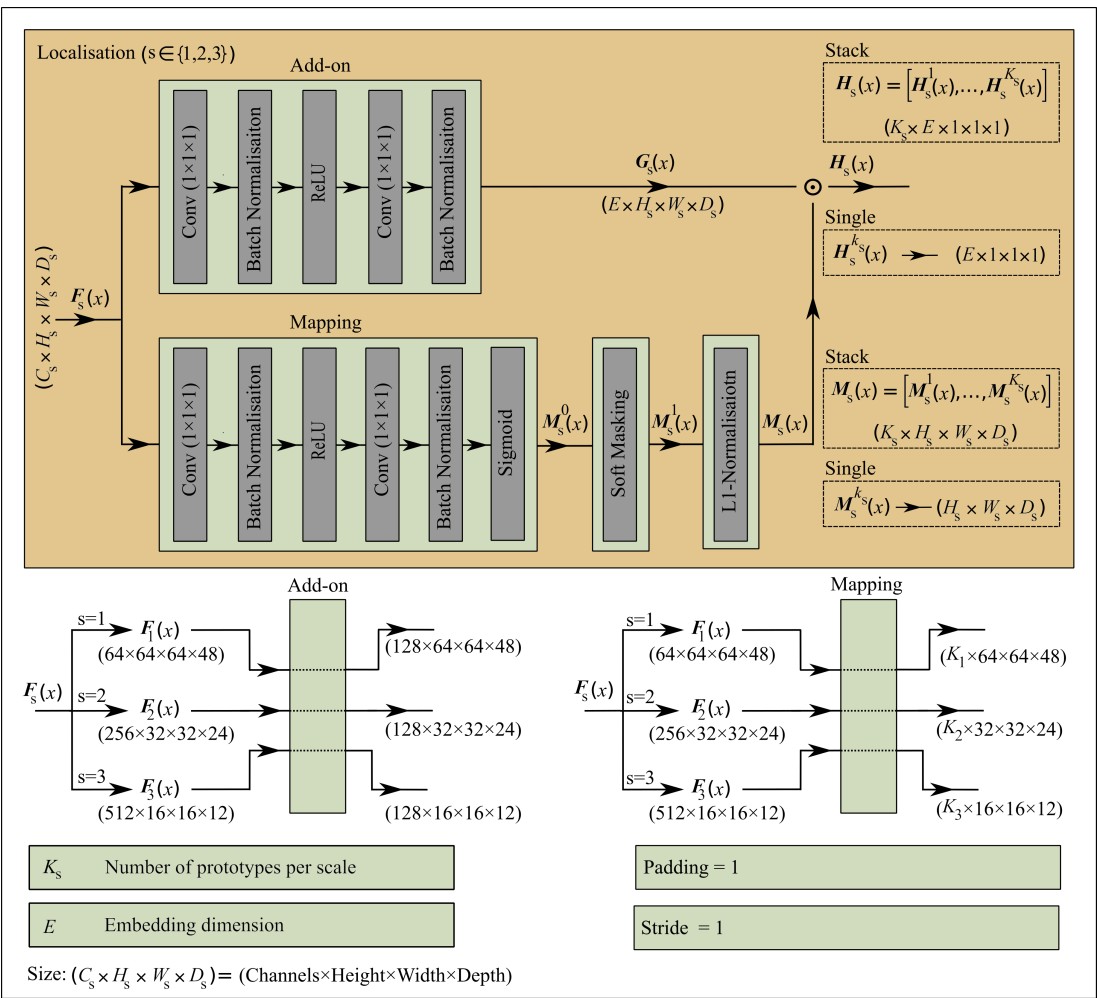

Figure 4: Localisation component.

**Prototype similarity**  At each scale, the cosine similarity between each prototype $\boldsymbol{p}_s^{k_s}$ and its corresponding feature descriptor $\boldsymbol{H}_s^{k_s}(x)$ is calculated,

$$S_s^{k_s}\left(x, \boldsymbol{p}_s^{k_s}\right) = \frac{\boldsymbol{H}_s^{k_s}(x) \cdot \boldsymbol{p}_s^{k_s}}{\left\|\boldsymbol{H}_s^{k_s}(x)\right\| \left\|\boldsymbol{p}_s^{k_s}\right\|}. \tag{16}$$

We then concatenate similarities across all prototypes and stack it into a single similarity vector $S(x, \boldsymbol{P}) \in \mathbb{R}^{K_{\text{total}}}$ to pass it to the classification component.

$$S(x, \boldsymbol{P}) = \left[S_1^1(x, \boldsymbol{p}_1^1), \ldots, S_1^{K_1}(x, \boldsymbol{p}_1^{K_1}), S_2^1(x, \boldsymbol{p}_2^1), \ldots, S_2^{K_2}(x, \boldsymbol{p}_2^{K_2}), S_3^1(x, \boldsymbol{p}_3^1), \ldots, S_3^{K_3}(x, \boldsymbol{p}_3^{K_3})\right]. \tag{17}$$

### A.4. Classification.

A linear classifier maps the similarity vector to class logits. To this end, similarity scores are multiplied to classifier weights $\boldsymbol{w}_{\text{cls}} \in \mathbb{R}^{C \times K_{\text{total}}}$ ($C$ is the number of classes). A softmax function then outputs the final predicted probabilities of $\boldsymbol{P}$.

$$\boldsymbol{y} = \text{softmax}\big(\boldsymbol{w}_{\text{cls}} \cdot S(x, \boldsymbol{P})\big). \tag{18}$$

## Appendix B. Training of UM–ProtoShare

### B.1. Training objectives

UM–ProtoShare is trained using the standard three-stage procedure adopted in case-based models (Chen et al., 2019; Wei et al., 2024):

**Stage (1) optimisation of components preceding the classification layer (Latent space).** In stage (1) we optimise all UM–ProtoShare layers up to and including the prototype bank. The goal is to obtain meaningful feature descriptors and prototype similarity patterns. The overall loss function for this stage is,

$$L_{\text{stage 1}} = \lambda_{\text{cls}} L_{\text{cls}} + \lambda_{\text{clst}} L_{\text{clst}}^{\text{class-agnostic}} + \lambda_{\text{sep}} L_{\text{sep}}^{\text{class-agnostic}} + \frac{\lambda_{\text{map}}}{3} \sum_{s=1}^{3} L_{\text{map}}^{s} + \frac{\lambda_{\text{OC}}}{3} \sum_{s=1}^{3} L_{\text{OC}}^{s} + \lambda_{\text{div}} L_{\text{div}}. \tag{19}$$

Where $\lambda_{\text{cls}}$, $\lambda_{\text{clst}}$, $\lambda_{\text{sep}}$, $\lambda_{\text{map}}$, $\lambda_{\text{OC}}$, and $\lambda_{\text{div}}$ are coefficients of respective loss terms. The classification loss $L_{\text{cls}}$ is the cross-entropy on class logits. The clustering loss $L_{\text{clst}}^{\text{class-agnostic}}$ and separation loss $L_{\text{sep}}^{\text{class-agnostic}}$ are formulated to be compatible with shared, class-agnostic prototypes, in contrast to class-specific design in UM–ProtoShare counterparts (Chen et al., 2019; Wei et al., 2024; Li et al., 2024), using soft prototype-class coefficients $\zeta_{c,\boldsymbol{p}_s^{k_s}}^{\text{soft}}$. These losses encourage high similarity to prototypes that support the ground-truth class and discourage similarity to prototypes that support other classes, respectively. The multi-scale mapping loss ($L_{\text{map}}^{s}$), and Online-CAM loss ($L_{\text{map}}^{s}$) follow the formulation of MProtoNet at each scale (Wei et al., 2024), and help localisation of prototype attention maps. The prototype-diversity regulariser $L_{\text{div}}$ penalises average pairwise cosine similarity between prototypes at each scale to reduce redundancy in the prototype bank.

**Stage (2) prototype reassignment ("push").** After the latent space has been optimised in Stage (1), we perform a prototype reassignment step to ensure that each prototype corresponds to a specific region of a training image. During this stage, all network parameters are frozen and, for each scale $s$, we replace the prototype $\boldsymbol{p}_s^{k_s}$ with its nearest feature descriptor $\boldsymbol{H}_s^{k_s}(x)$ in the latent space at the same scale. Let $X_N = \{x_n \mid n \in \{1, \ldots, N\}\}$ denote the training images, and for each image $x_n$, $\boldsymbol{H}_s^{k_s}(x_n)$ denote the corresponding feature descriptors. We then search over all training images for each prototype $\boldsymbol{p}_s^{k_s}$,

$$\boldsymbol{p}_s^{k_s} \leftarrow \boldsymbol{H}_s^{k_s}(x_{n'}), \quad n' = \arg \max_{1 \leq n \leq N} S_s^{k_s}\big(x_n, \boldsymbol{p}_s^{k_s}\big). \tag{20}$$

Here $S_s^{k_s}(x_n, \boldsymbol{p}_s^{k_s})$ denote the similarity score between prototype $\boldsymbol{p}_s^{k_s}$ and its corresponding feature descriptor $\boldsymbol{H}_s^{k_s}(x_n)$ for image $x_n$. This "push" operation anchors each prototype to a specific latent descriptor from a real training image, enabling faithful visualisation via its attention map $\boldsymbol{M}_s^{k_s}(x_n)$.

**Stage (3) training of the classification component**  After prototype reassignment, we train the classification layer separately to learn weight coefficients $\boldsymbol{w}_{\text{cls}} \in \mathbb{R}^{C \times K_{\text{total}}}$, that represent the contributions of prototypes to classification without changing their representations. The overall loss at this stage is,

$$L_{\text{stage 3}} = \lambda_{\text{cls}} L_{\text{cls}} + \lambda_{L_1} L_{L_1}. \tag{21}$$

Where $L_{L_1}$ is the L1-regularisation term which is originated in the classification component of ProtoPNet (Chen et al., 2019). $\lambda_{\text{cls}}$, and $\lambda_{L_1}$ are coefficients for respective losses.

### B.2. Clustering and separation losses: class-specific formulation (ProtoPNet)

In ProtoPNet–style models, each prototype is assigned to exactly one class through an ownership matrix $\boldsymbol{A} \in \{0,1\}^{C \times K_{\text{total}}}$, where $j \in \{1, \ldots, K_{\text{total}}\}$ indexes prototypes. The entry $A_{c,j} = 1$ if prototype $j$ belongs to class $c$ and $A_{c,j} = 0$ otherwise, with $\sum_{c=1}^{C} A_{c,j} = 1$ for every prototype index $j$ (each $j$ corresponds to one prototype $\boldsymbol{p}_s^{k_s}$). The resulting hard class–prototype coefficient is $\zeta_{c,\boldsymbol{p}_s^{k_s}}^{\text{hard}} = A_{c,j} \in \{0,1\}$, meaning that prototype $\boldsymbol{p}_s^{k_s}$ supports class $c$ if $\zeta_{c,\boldsymbol{p}_s^{k_s}}^{\text{hard}} = 1$ and does not support it otherwise. Using the similarity scores $S_s^{k_s}(x, \boldsymbol{p}_s^{k_s})$, the class-specific clustering and separation losses for an input $(x, y = c)$ across scales are (Chen et al., 2019),

$$L_{\text{clst}}^{\text{class-specific}} = -\max_{s,k_s}\left( \zeta_{c,\boldsymbol{p}_s^{k_s}}^{\text{hard}} \, S_s^{k_s}(x, \boldsymbol{p}_s^{k_s}) \right). \tag{22}$$

$$L_{\text{sep}}^{\text{class-specific}} = \max_{s,k_s}\left( \left(1 - \zeta_{c,\boldsymbol{p}_s^{k_s}}^{\text{hard}}\right) S_s^{k_s}(x, \boldsymbol{p}_s^{k_s}) \right). \tag{23}$$

## Appendix C. Preprocessing and data augmentation

For each patient, all MRI sequences were co-registered to the SRI24 anatomical template (Rohlfing et al., 2010), resampled to $1.5\,\text{mm}^3$ isotropic voxels, and cropped to $128 \times 128 \times 96$ centred on the brain. Z-score normalisation was applied on each sequence. To mitigate data scarcity, stochastic data augmentation was applied during training, including (i) rotation and scaling, (ii) Gaussian noise, (iii) Gaussian blur, (iv) brightness adjustment, (v) contrast adjustment, (vi) simulation of low resolution, (vii) gamma augmentation, and (viii) mirroring. Detailed parameters for each augmentation operation are provided in (Isensee et al., 2021). To ensure fair comparison with prior case-based models, we followed the same preprocessing and augmentation pipelines as in (Wei et al., 2024; Li et al., 2024).

## Appendix D. Evaluation metrics and training hyper-parameters

### D.1. Evaluation metrics

We evaluate UM–ProtoShare in terms of both classification performance and interpretability. Classification performance is assessed with Balanced Accuracy (BAC). Interpretability is assessed in terms of localisation coherence and correctness (Nauta et al., 2023), using Incremental Deletion Score (IDS) and Activation Precision (AP), respectively.

### D.2. Balanced accuracy (BAC)

To mitigate the LGG/HGG class imbalance, we report Balanced Accuracy (BAC), defined as the mean of sensitivity and specificity,

$$\text{BAC} = \frac{\text{TPR} + \text{TNR}}{2}. \tag{24}$$

Where TPR is the true positive rate (sensitivity) and TNR is the true negative rate (specificity). This gives equal weight to each class and mitigates the impact of class imbalance.

### D.3. Activation precision (AP)

Activation Precision (AP) measures the localisation coherence of the attribution maps by quantifying how well they align with the human-annotated tumour masks. Let $\boldsymbol{M}(x)$ denote the attention map for an input image $x$ (The construction of $\boldsymbol{M}(x)$ is described in Appendix F) and let $\boldsymbol{W}_T(x)$ be the corresponding human-annotated tumour mask. We first upsample $\boldsymbol{M}(x)$ to the input space using trilinear interpolation, denoted by UpSample($\boldsymbol{M}(x)$). A binary threshold function $T(\cdot)$ with threshold 0.5 (following (Barnett et al., 2021)) is then applied. AP is defined as,

$$\text{AP} = \frac{|\boldsymbol{W}_T(x) \cap T(\text{UpSample}(\boldsymbol{M}(x)))|}{|T(\text{UpSample}(\boldsymbol{M}(x)))|}. \tag{25}$$

### D.4. Incremental deletion score (IDS)

Incremental Deletion Score (IDS) measures the correctness and the faithfulness of the attribution maps to the model's decision. Given upsampled attribution map $\boldsymbol{M}(x)$ voxels are ranked in descending order of their attribution scores. A fixed fraction of the most highly attributed voxels is progressively removed (their intensities are replaced with 0), and the model's classification accuracy is recorded after each deletion step, resulting in an accuracy–deletion curve. IDS is defined as the normalised area under the accuracy-deletion curve over the specified start–end bounds. Lower IDS indicates that activation maps are truly critical for model reasoning (deletion causes a rapid accuracy drop) and thus indicates more faithful attribution (Nauta et al., 2023).

### D.5. Hyper-parameters during training

Following the training practice in (Wei et al., 2024), we train UM–ProtoShare using 5-fold cross-validation as described in the main text. For each cross-validation fold, we first warm up the backbone for 50 epochs using AdamW optimiser with a learning rate of $10^{-3}$, weight decay of $10^{-2}$, and a batch size of 32. In this warm-up stage, and following the strategy explained in the main text, we first initialise the encoder features and then train the whole backbone while jointly training the encoder, decoder, and gated fusion modules. Using the same training parameters, we then train UM–ProtoShare for 100 epochs while the decoder is frozen. This preserves the locality benefits of the decoder while keeping training efficiency comparable to prior case-based models. During these 100 epochs we apply a learning-rate scheduler with linear warm-up for the first 20 epochs, followed by cosine annealing for the remaining 80 epochs. After every 10 epochs of Stage (1), we execute Stage (2) (prototype

reassignment to the closest feature descriptors), followed by Stage (3) for 10 epochs, during which the classification layer is trained with the Adam optimiser at a learning rate of $10^{-3}$. Data augmentation described in Appendix C is not applied during Stage (2). To ensure reproducibility and avoid over-tuning, we follow the standard loss-weighting practice in prior case-based baselines (Wei et al., 2024) and keep all coefficients fixed across cross-validation folds, ablations, and comparative experiments. Specifically, the loss coefficients are $\lambda_{\text{cls}} = 1$, $\lambda_{\text{clst}} = 0.8$, $\lambda_{\text{sep}} = 0.08$, $\lambda_{\text{map}} = 0.5$, $\lambda_{\text{OC}} = 0.05$, $\lambda_{\text{div}} = 0.001$, and $\lambda_{L_1} = 0.01$. All models were trained on a single NVIDIA A100 GPU.

## Appendix E. Ablation study configurations

Concretely, A1-A2 compare class-specific and class-agnostic prototype banks in a single-scale, encoder-only setting; B introduces multi-scale prototypes without using the decoder; C enables the decoder with simple channel-wise concatenation; D replaces concatenation with gated fusions; and E varies the allocation of prototypes across scales while keeping the total prototype budget fixed. The total prototype budget is fixed at $K_{\text{total}} = 30$. For multi-scale variants (Ablations B-D), we use a balanced prototype allocation across scales $(K_1, K_2, K_3) = (10, 10, 10)$. Ablation E evaluates the effect of per-scale allocation (fine, mid, coarse). At each ablation the effects on the classification performance (BAC) and interpretability (IDS, AP) are investigated.

### E.1. Ablation A (class-specific prototypes (A1) versus class-agnostic prototypes (A2))

To isolate the effect of prototype sharing, we disable the decoder, all gated fusions, and use only the deepest encoder features $\boldsymbol{f}_3(x)$. The prototype bank is restricted to a single scale with $K_3 = 30$. We then compare two configurations:

- **Ablation A1 (class-specific):** Two distinct slots of 15 prototypes are assigned to the two classes via a hard class–prototype ownership matrix. This corresponds to MProtoNet with a normalised mapping module (see Equation 2 in the main text). This configuration (A1) mirrors the MProtoNet design but adds normalised mapping head.

- **Ablation A2 (class-agnostic):** All 30 prototypes are shared across classes and may support either class through the soft prototype-class coefficients $\zeta^{\text{soft}}_{c,\boldsymbol{p}_s^{k_s}}$ defined in Section 2.3

### E.2. Ablation B (multi-scale prototypes, no decoder)

To assess the effect of explicit multi-scale prototypes, we use the three encoder features $\{\boldsymbol{f}_1(x), \boldsymbol{f}_2(x), \boldsymbol{f}_3(x)\}$, and train separate prototype sets per scale with $(K_1, K_2, K_3) = (10, 10, 10)$ so that $K_{\text{total}} = 30$. The decoder and all gated fusions remain disabled, and the prototypes are shared and class-agnostic at all scales. Any differences relative to A2 can therefore be attributed to introducing multi-scale prototypes.

### E.3. Ablation C (decoder effect)

In ablation C we enable the decoder while keeping the shared, class-agnostic, multi-scale prototype bank unchanged. Encoder and decoder feature maps at each scale are fused by channel-wise concatenation, followed by a linear projection ($1 \times 1 \times 1$ convolution) to restore the fused channels to $C_s$. All gated fusions remain disabled. This configuration isolates the contribution of the decoder to feature quality and prototype localisation, without the confounding effect of gated fusion.

### E.4. Ablation D (gated fusion effect)

In ablation D we replace the simple channel-wise concatenation fusion in Ablation C with gated fusions, allowing the model to learn the relative contribution of encoder and decoder features at each scale (see Equation (11) in Appendix A.1). Comparing D against C isolates the effect of gated fusions on the classification performance and interpretability quality.

### E.5. Ablation E (prototype scale allocation)

While using a balanced prototype allocation across scales, providing capacity to capture tumour features from fine to coarse scales while preserving a balanced contribution from each scale $(K_1, K_2, K_3) = (10, 10, 10)$, in ablation $E$ we investigate how per-scale prototype allocation affects classification performance (BAC) and interpretability (AP, IDS) under a fixed total budget of $K_{\text{total}} = 30$. We consider three allocation schemes:

- **Ablation E1 (Fine-heavy):** $(K_1, K_2, K_3) = (15, 9, 6)$
- **Ablation E2 (Mid-heavy):** $(K_1, K_2, K_3) = (9, 15, 6)$
- **Ablation E3 (Coarse-heavy):** $(K_1, K_2, K_3) = (6, 9, 15)$

All other architectural and training settings are kept identical across these configurations.

## Appendix F. Comparative benchmark configurations

We compare UM–ProtoShare against a CNN model derived from the UM–ProtoShare backbone, ProtoPNet (Chen et al., 2019), XProtoNet (Kim et al., 2021), MProtoNet (Wei et al., 2024), and MAProtoNet (Li et al., 2024). These models are selected because they represent recent state-of-the-art case-based models that have been adapted for clinical imaging applications.

For a fair comparison, all models use identical data splits at each cross-validation fold, MRI preprocessing and augmentation, training schedules, and a prototype bank with the same total number of prototypes; only the architectural design differences are varied. ProtoPNet, XProtoNet and MProtoNet are re-implemented within the UM–ProtoShare codebase to ensure consistent architecture components and training routines. For MAProtoNet, we use the public implementation and align the preprocessing and training procedure to match the other models. As all benchmark models use single-scale prototypes at the deepest backbone features, their prototype and attribution terms are written with subscript $s = 3$, for consistency with the UM–ProtoShare notation.

### F.1. Model design

**CNN model.** The CNN baseline uses the same encoder–decoder backbone as UM–ProtoShare. It retains the add-on module used in the localisation component (Figure 1) and connects it directly to the classification layer via global average pooling.

**ProtoPNet.** ProtoPNet is adapted to multi-sequence 3D MRI with a single-scale, class-specific prototype bank. Prototype similarities $S_3^{k_3}(x, \boldsymbol{p}_3^{k_3})$ are computed by directly comparing the latent features $\boldsymbol{G}_3(x)$ with each prototype $\boldsymbol{p}_3^{k_3}$ using the top-$\alpha$ average pooling ($\alpha = 1\%$), reflecting the absence of mapping module in its design.

**XProtoNet.** XProtoNet is likewise adapted to the same multi-sequence 3D MRI input. In contrast to ProtoPNet, it includes a mapping module that produces prototype attention maps, from which class-specific prototype similarities $S_3^{k_3}(x, \boldsymbol{p}_3^{k_3})$ are obtained.

**MProtoNet.** MProtoNet extends XProtoNet by introducing soft masking and an Online-CAM module, which further refines attribution maps and logits. Prototypes are still single-scale and class-specific, but the attribution maps are influenced by the Online-CAM-based pooling.

**MAProtoNet.** MAProtoNet further employs multi-scale feature fusion and a multi-scale mapping loss, while its prototype bank remains single-scale and class-specific. Multi-scale features are fused before computing prototype similarities, whereas UM–ProtoShare uses a multi-scale, shared, class-agnostic prototype bank.

### F.2. Attention map construction for AP and IDS

For the interpretability metrics (AP and IDS), we require an attention map $\boldsymbol{M}(x)$ for each model (see Appendix D).

**CNN model (Grad-CAM).** For the CNN baseline, $\boldsymbol{M}(x)$ is given by the gradient-based Grad-CAM (Selvaraju et al., 2017) map computed from the last convolutional layer, highlighting regions that most support the class of interest.

**CNN model (Occlusion).** We also include Occlusion (Zeiler and Fergus, 2014) as a perturbation-based alternative to Grad-CAM. Here $\boldsymbol{M}(x)$ is given by the Occlusion sensitivity map, computed by sliding a window of size 11 over the input volume with stride 5 (in each spatial dimension) and measuring the change in the target class score when each region is replaced with zeros.

**ProtoPNet.** For ProtoPNet, there is no explicit mapping module. We therefore use the similarity scores as attribution: for each prototype we take the similarity scores over spatial locations (patches of $\boldsymbol{G}_3(x)$ with the same size of prototypes), select the top-$\alpha$ most activated patches ($\alpha = 1\%$), and average their scores to form an attention map. Class-specific maps are obtained by aggregating only prototypes assigned to the corresponding class $\boldsymbol{M}(x)$.

**XProtoNet, MProtoNet and MAProtoNet.** For XProtoNet, MProtoNet and MAProtoNet, we construct the attention map for class $c$ by averaging the attention maps of all

prototypes assigned to that class. Let $K_{3,c}$ denote the total number of prototypes assigned to class $c$, and $\boldsymbol{M}_{3,c}^{k_3}(x)$ be the attention map of prototype $\boldsymbol{p}_3^{k_3}$ for class $c$. Then,

$$\boldsymbol{M}(x) = \frac{1}{K_{3,c}} \sum_{k_3=1}^{K_{3,c}} \boldsymbol{M}_{3,c}^{k_3}(x). \tag{26}$$

These attribution maps are then used to compute AP and IDS as described in Appendix D.

**UM–ProtoShare.** In UM–ProtoShare, prototypes are shared across classes and their class-specific support is quantified by soft class–prototype coefficients $\zeta_{c,\boldsymbol{p}_s^{k_s}}^{\text{soft}}$, defined in the main text. So, if and only if $\zeta_{c,\boldsymbol{p}_s^{k_s}}^{\text{soft}} > 0$ the ReLU-activated classifier weight for class $c$ and prototype $\boldsymbol{p}_s^{k_s}$ is positive (see Equation (7)). We therefore define the set of prototypes that support class $c$ at scale $s$,

$$K_{(s,c)}^{(+)} = \left\{ \boldsymbol{p}_s^{k_s} \mid \zeta_{c,\boldsymbol{p}_s^{k_s}}^{\text{soft}} > 0 \right\}. \tag{27}$$

Let $\boldsymbol{M}_{c,s}^{k_s}(x)$ denote the attention map of prototype $\boldsymbol{p}_s^{k_s}$ for input $x$ at scale $s$. The attention map $\boldsymbol{M}(x)$ for UM–ProtoShare is then obtained by averaging the attention maps of prototypes that support class $c$ across all scales,

$$\boldsymbol{M}(x) = \sum_{s=1}^{3} \frac{1}{|K_{(s,c)}^{(+)}|} \sum_{\boldsymbol{p}_s^{k_s} \in K_{(s,c)}^{(+)}} \boldsymbol{M}_{c,s}^{k_s}(x). \tag{28}$$

These class-specific maps are then used to compute AP and IDS as described in Appendix D.

## Appendix G. Qualitative interpretability analysis

Figure 5 provides a qualitative, clinically grounded view of how UM–ProtoShare supports its predictions by reusing shared class-agnostic prototypes across grades, aggregating evidence across fine, mid, and coarse scales, and integrating complementary cues from multi-sequence 3D MRI. We explicitly separate pattern reusability (shared prototype similarity across grades) from diagnostic effect (class-specific contribution through the classifier weights), which is particularly relevant in clinical practice where tumour-related MRI cues may overlap across grades and appear ambiguous in isolation without multi-sequence context. We use the E2 configuration as the default throughout this analysis.

### G.1. Framework

We select two representative cases from different classes, one LGG case $x^{\text{LGG}}$ and one HGG case $x^{\text{HGG}}$. Prototypes are re-learned per cross-validation fold and may differ in exact evidence patches due to the push-based anchoring to training samples. For each scale $s \in \{1, 2, 3\}$, UM–ProtoShare computes prototype similarities $S_s^{k_s}(x, \boldsymbol{p}_s^{k_s}) = \cos\!\left(\boldsymbol{H}_s^{k_s}(x), \boldsymbol{p}_s^{k_s}\right)$ for all prototypes $\{\boldsymbol{p}_s^{k_s}\}_{k_s=1}^{K_s}$. To identify prototypes that are strongly activated in both classes, we define a sharedness score for each prototype as the minimum similarity across the two cases,

$$\phi_s^{k_s} = \min\Big( S_s^{k_s}\big(x^{\text{LGG}}, \boldsymbol{p}_s^{k_s}\big), S_s^{k_s}\big(x^{\text{HGG}}, \boldsymbol{p}_s^{k_s}\big)\Big). \tag{29}$$

This criterion ensures that selected prototypes have high similarity for both cases, rather than being highly activated in only one class. At each scale $s$, we then select the most shared prototype as,

$$k_s^* = \arg \max_{1 \le k_s \le K_s} \phi_s^{k_s}. \tag{30}$$

For the selected shared prototype $\boldsymbol{p}_s^{k_s^*}$, we visualise (i) the corresponding attention maps for both $x^{\text{LGG}}$ and $x^{\text{HGG}}$, and (ii) the prototype evidence patch associated with $\boldsymbol{p}_s^{k_s^*}$ obtained after the prototype reassignment (push) stage. Finally, to explicitly connect similarity-based matching to the classification decision, we report the per-class contribution of this prototype to the class logit, computed as,

$$\Delta y_c^{(s,k_s^*)}(x) = w_{\text{cls}}[c, s, k_s^*]\, S_s^{k_s^*}(x, \boldsymbol{p}_s^{k_s^*}). \tag{31}$$

Where $w_{\text{cls}}[c, s, k_s^*]$ denotes the classifier weight linking prototype $\boldsymbol{p}_s^{k_s^*}$ to class $c$. This analysis demonstrates that the same shared prototype can strongly match tumour-related regions in both LGG and HGG cases (high similarity), while its influence on each class prediction is modulated by the class-specific weights, yielding positive evidence or counter-evidence depending on the target class.

### G.2. Interpretation and clinical implication

The examples in Figure 5 illustrate that prototypes may capture tumour-related patterns shared across grades, while their diagnostic role is determined by class-specific classifier weights, such that the same matched evidence can act as supporting evidence for one class and counter-evidence for the other.

**Scale 1 (fine-scale) prototype interpretation.** The selected prototype is anchored to an HGG training patch, yet it achieves high similarity for both the LGG and HGG cases, indicating a fine-scale appearance pattern shared across grades. Although the LGG case matches this prototype strongly in T1-CE, its contribution to the LGG logit is negative, meaning that this shared cue is treated as counter-evidence for LGG and more consistent with HGG. This behaviour is clinically plausible, since local enhancement-like patterns may occur in both grades and can be ambiguous in isolation. Complementary evidence in other sequences (e.g., oedema extent and tissue heterogeneity) provides additional context that helps disambiguate grade and stabilise the decision.

**Scale 2 (mid-scale) prototype interpretation.** The selected mid-scale prototype is anchored to an LGG training patch and achieves high similarity for the LGG case, suggesting a recurring mid-scale pattern that is characteristic of lower-grade tumours. This similarity is not uniformly driven by all sequences: while the T1-CE appearance differs, the match becomes more evident in T1 and FLAIR (and to a lesser extent T2), indicating a multi-sequence signature rather than a single-modality cue. Notably, the HGG case also exhibits a relatively strong match in T2, highlighting that single-sequence interpretation could be

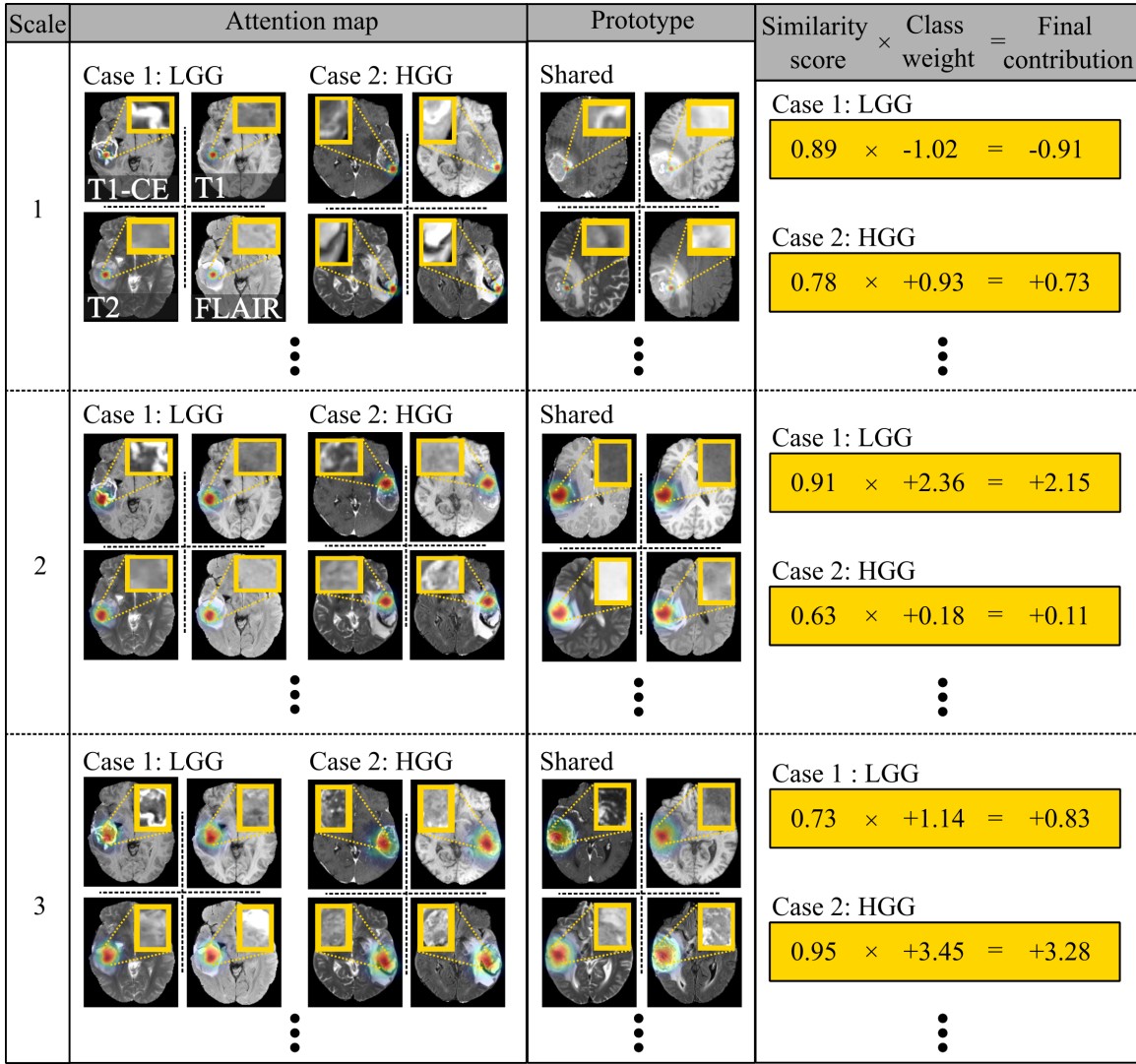

Figure 5: Examples of shared prototype activation across LGG/HGG cases at fine/mid/coarse scales, showing prototype evidence patches, attention maps, and per-class contributions.

misleading. Consistent with its mid-scale role, the prototype contributes more positively to the LGG logit, reflecting sensitivity to broader structural context (e.g., boundary coherence and tissue homogeneity across sequences).

**Scale 3 (coarse-scale) prototype interpretation.** The selected coarse-scale prototype is anchored to an HGG training case and captures a global tumour-level pattern expressed consistently across modalities. In the HGG case, the match is strong in all sequences, with particularly clear correspondence in T1-CE and reinforced by coherent patterns in T1, T2

and FLAIR. In contrast, the LGG case differs more noticeably in T1-CE and FLAIR, while partial similarity emerges mainly through T1 and T2, indicating that some global tumour characteristics can overlap across grades. Clinically, this reflects that coarse-scale evidence (e.g., global heterogeneity and lesion configuration) becomes most indicative of high-grade pathology when it persists across modalities.

## Appendix H. Extended discussion

UM–ProtoShare is a case-based model that combines a 3D ResNet-152 encoder, a UNet–style decoder with gated fusions, and a bank of shared, class-agnostic, multi-scale prototypes for brain tumour classification from multi-sequence 3D MRI. By integrating transfer learning through a pre-trained 3D encoder, and by jointly optimising a normalised mapping module, class-agnostic prototype sharing and explicit multi-scale prototype banks, UM–ProtoShare aims to mitigate the conventional accuracy–interpretability trade-off in case-based models. The ablation study shows that these components collectively improve both BAC and interpretability metrics (AP, IDS) relative to prior case-based models, while remaining competitive with a strong CNN model.

Normalising ($\ell_1$-normalisation) the mapping module helps reduce tumour size effects in the attention maps. Moving from MProtoNet to the A1 variant, which mirrors the MProtoNet architecture but uses the normalised mapping, yields modest gains in BAC and AP and a reduction in IDS, consistent with the goal of attenuating size bias and producing cleaner evidence maps. Normalisation also makes attention values more comparable across different tumour volumes, so evidence reflects where the model focuses rather than how large the lesion is, in line with recommendations from weakly supervised localisation methods where normalised attention improves localisation coherence (Xu et al., 2022). Adding a UNet–style decoder recovers spatial resolution and fine structure attenuated by encoder downsampling, a common strategy in medical imaging to sharpen boundaries and improve localisation over encoder-only designs (Oktay et al., 2018; Navab et al., 2015). In our ablation (B→C), the decoder increases AP and reduces IDS at a modest cost in BAC, suggesting more precise attention maps under weak supervision (Oktay et al., 2018). Replacing simple concatenation with encoder–decoder gated fusions (C→D) then regulates how features contribute at each scale, recovering most of the lost accuracy while maintaining low IDS and high AP, and yielding clearer, scale-coherent maps (Oktay et al., 2018; Hu et al., 2018; Takikawa et al., 2019). Under a fixed prototype budget ($K_{\text{total}} = 30$), the E-variants illustrate a scale-dependent accuracy–interpretability trade-off: fine-heavy allocation emphasises edge and boundary cues (e.g., thin enhancing rims, irregular margins), maximising AP and minimising IDS; coarse-heavy allocation emphasises global context (oedema extent, mass effect), achieving the highest BAC; and the mid-heavy configuration (E2) provides the most balanced operating point.

Class-agnostic prototype sharing and an explicit multi-scale prototype bank are the central architectural choices that distinguish UM–ProtoShare from prior case-based models. Shared, class-agnostic prototypes allow the model to represent appearance patterns that occur in both tumour grades, which is realistic for multi-sequence MRI where grades often differ by intensity and extent rather than entirely distinct morphologies (Haydar et al., 2022). Sharing also reduces redundancy in the prototype space and improves data

efficiency, as high-quality prototypes can be reused across classes instead of duplicated (Rymarczyk et al., 2021). In the ablation, moving from class-specific (A1) to class-agnostic (A2) prototypes increases BAC and AP with only a mild change in IDS, and qualitatively provides broader coverage across MRI sequences, particularly in mixed or borderline cases. A potential risk is over-sharing, where prototypes drift towards the majority class; in UM–ProtoShare this is mitigated by soft, class-conditioned assignment in the separation loss and by a diversity regulariser (Gautam et al., 2023). Overall, shared prototypes improve coverage of relevant tumour features and contribute to the observed gains in interpretability without sacrificing classification performance or increasing the number of prototypes.

The multi-scale prototype bank enables attention from fine- to coarse-grained regions in MRI. Smaller-scale prototypes sharpen localisation, whereas larger-scale prototypes preserve global anatomical context and reduce sensitivity to small spurious textures, a known issue for case-based models (Lu et al., 2025), thereby helping to balance accuracy and interpretability (Li et al., 2024). This design is well matched to the multi-scale imaging characteristics of brain tumours, where fine-grained cues (rim or irregular enhancement, small non-enhancing components, irregular margins) coexist with coarse-grained cues (oedema extent, mass effect, cross-sequence concordance) (Wegscheid and Jennings, 2024; Martucci et al., 2023). The approach also aligns with radiomics, which routinely applies multi-scale filters (e.g., wavelets, Laplacian of Gaussian) (Golbaf et al., 2025), and has been standardised by the Image Biomarker Standardisation Initiative (IBSI) (Zwanenburg et al., 2020). Consistently, our ablation (A2→B) shows that introducing an explicit multi-scale prototype bank improves both BAC and interpretability metrics.

Regarding multi-scale capabilities, the closest counterparts are MAProtoNet and HQProtoPNet (Wang et al., 2023). MAProtoNet fuses multi-scale encoder features before the mapping module and supervises them with a multi-scale mapping loss, but its prototypes remain single-scale and class-specific. HQProtoPNet retains single-scale, class-specific prototypes and attains multi-scale behaviour via a multi-scale matching layer that reshapes backbone features into multiple pooled grids, so the same prototype matches across scales without distinct per-scale banks. By contrast, UM–ProtoShare learns explicit prototype banks at fine, mid and coarse scales and enhances locality through a UNet–style decoder with gated fusions. In terms of prototype sharing, the closest methods are ProtoPool (Rymarczyk et al., 2022), ProtoPShare (Rymarczyk et al., 2021) and structurally ProtoTree (Nauta et al., 2021). ProtoPShare introduces sharing post-hoc by merge-and-prune, ProtoTree shares prototypes implicitly via decision-tree routing, and ProtoPool maintains a global prototype pool with differentiable Gumbel–softmax slot assignment. UM–ProtoShare instead learns shared prototypes using normalised soft prototype–class coefficients derived from Grad-CAM–style importance weights, so that prototypes can support multiple classes while clustering and separation losses are modulated accordingly. This mechanism integrates naturally with the mapping module and Online-CAM used to supervise prototype attention maps, yielding a unified framework for multi-scale, class-agnostic, case-based explanations in multi-sequence 3D MRI.

