# OpenReview forum: "UM--ProtoShare: UNet-Guided, Multi-scale Shared Prototypes for Interpretable Brain Tumour Classification Using Multi-sequence 3D MRI"
_MIDL.io/2026/Conference — MIDL 2026 Poster_

### Official Review · Reviewer_JLd3 · 2026-01-02

**Confidence:** 3
**Preliminary Rating:** 4
**Final Rating:** 4

**Summary:**

This paper proposes UM-ProtoShare, a case-based/prototype model for binary glioma grading (LGG vs. HGG) from multi-sequence 3D MRI. UM-ProtoShare embeds interpretability into the decision process to increase the explainability of the machine learning model.
It learns a bank of shared, class-agnostic prototypes distributed across different scales, and its backbone architecture is a 3D encoder and a lightweight UNet-style decoder. A normalized soft-masked mapping module is used to align prototype evidence.
In the result section, the evaluation is reported via 5-fold cross-validation on BraTS-2020 (369 glioma cases) with Balanced Accuracy (BAC) and interpretability metrics, Activation Precision (AP), and Incremental Deletion Score (IDS). The method attains competitive balanced accuracy while increasing the interpretability scores.

**Strengths:**

The paper introduces a multi-sequence case-based method on MRI data, which fulfills the gap of interpretable multi-sequence models.
The methodology is well-written and structured.
The evaluation includes the baseline comparison with prototype-based models and clean ablation ladder, which shows the contribution of the work. In the table 1, the A1→E variants isolate effects of sharing, multi-scale prototypes, decoder, and gated fusion under a fixed number of prototypes.
Furthermore, the quantitative interpretability evaluation is reasonable,  AP/IDS are computed consistently across models and the deletion-based faithfulness metric is a reasonable complement to localisation overlap.
The soft ownership mechanism is interesting. Deriving ζ from positive classifier weights lets prototypes support multiple classes while still shaping separation.

**Weaknesses:**

The authors evaluate the proposed method on a single dataset: BraTS-2020 and use it as a binary task, while the authors acknowledge the need for larger cohorts and external validation.
The motivation of the work operates on the multi-sequence/modality MRI volumes. However, the authors do not compare with single modalities, whether the prototypes would change and also the accuracy of the models. It is difficult to say which modality supports which prototype match.
The qualitative evaluation of the multi-sequence prototypes is missing. Whether the prototypes stay constantly, when the models are trained with different folds/seeds? Whether the prototyps match between different modalities?

**Detailed Comments:**

- The authors might need to add a dataset to support the claim in the methodology.
- Computation power: The authors could provide the computation power they used during training, since the models are trained on 3D MRI data.
- For the reproducibility: The GitHub link does not work.
- In the evaluation part, the reviewers/readers need to continuously look into the supplementary to understand the experimental setting. It would be nicer if section E. could be explained in the main text.

**Justification Of Final Rating:**

The authors added additional descriptions and answered my concerns during the rebuttal phase. This increases the readability of the manuscript. I would like to maintain the current rating "weak accept".

**Justification Of The Preliminary Rating:**

The paper is well-written and motivated, and fills the gap in the multi-sequence interpretability of machine learning models. The evaluation is well defined by comparing with other prototype-based models (ProtoPNET, XProtoNet, MProtoNet, MAProtoNet). In the ablation study, the authors provide more insights into the model performance with different design configurations, and the results are competitive with the baselines and increase the interpretability when it comes to segmenting tumours. Nevertheless, the model could be further evaluated in an external dataset or with more complex multi-class tasks.

**Questions To Address In The Rebuttal:**

- Prototype stability: Do the prototypes change across different folds of validation? Is there any qualitative analysis on the consistency of the prototypes, and also across multiple sequence data?

- Do the authors try on different tasks, such as multi-classification tasks?

- For CNN baselines, Grad-CAM performs quite badly compared with other baseline methods. Could the authors provide some other different post-hoc methods to justify the consistency of the result?

---

> ### Author Response · Authors · 2026-01-24
>
> We thank the reviewer for their careful evaluation and constructive feedback. We have revised the manuscript accordingly, and provide point-by-point responses below (changes are highlighted in yellow in the revised manuscript).
>
> \textbf{Weakness:}
> We agree that evaluating on a single dataset (BraTS-2020) and framing the task as binary LGG vs HGG limits the scope of validation, and that broader validation would further strengthen generalisability. In this work, we follow the established BraTS-2020 multi-sequence 3D MRI setting used by recent case-based baselines to enable a controlled, capacity-matched comparison, and we highlight evaluation on additional datasets/tasks as future work (Limitations and Future Work, Page 12).
>
> We also agree that, because UM--ProtoShare learns prototypes from stacked multi-sequence inputs, it is not always trivial to attribute a prototype match to a single MRI sequence in isolation. To address the missing qualitative single/multi-sequence perspective, we added a dedicated Qualitative interpretability analysis section (Appendix G, Page 29; Figure 3, Page 31) showing how shared prototypes are reused across grades and how single/multi-sequence evidence contributes across fine/mid/coarse scales.
>
> Regarding stability across folds/seeds, we report mean $\pm$ std over 5-fold cross-validation for all metrics (Table 1, Page 10), and the consistent ablation/benchmark trends suggest robustness across splits; a dedicated analysis of modality-specific prototype alignment and prototype stability across folds/seeds is highlighted as future work (Limitations and Future Work, Page 12).
>
> \textbf{Detailed comment 1:} We agree that evaluating UM--ProtoShare on additional datasets would further strengthen the generalisability of the methodology and support broader claims. In this work, we focus on BraTS-2020 as a clinically relevant multi-sequence 3D MRI benchmark, and follow the established setting used by recent case-based baselines for fair comparison. Importantly, the shared class-agnostic multi-scale prototype formulation and training objectives are not dataset-specific and can be applied to other medical image classification tasks. This limitation and evaluation on additional datasets/tasks are highlighted as future work (Limitations and Future Work, Page 12).
>
> \textbf{Detailed comment 2:} We added GPU hardware details (single NVIDIA A100) (Appendix D.5, Page 26).
>
> \textbf{Detailed comment 3:} The repository is now public, and the GitHub link has been verified to be fully operational.
>
> \textbf{Detailed comment 4:} We agree that placing experimental details in the appendix can reduce readability. We added a short explanation of each experimental protocol at the start of the Ablation studies, while keeping Appendix E (Page 26) for full reproducibility. We also strengthened the Results narrative in the Ablation studies (Pages 9--10) and Comparative benchmark (Pages 10--11) to reduce reliance on the appendix for the primary discussion.
>
> \textbf{Question 1:} In our 5-fold cross-validation setting, prototypes are re-learned independently within each fold; since the training set changes and the push stage anchors each prototype to a real training patch, the exact evidence patches are not expected to be identical across folds. Therefore, a strict prototype consistency analysis across folds is not directly applicable. Nevertheless, prototype behaviour and localisation trends remain consistent across folds, as reflected by mean $\pm$ std performance over cross-validation (particularly for AP/IDS). In addition, we added a qualitative interpretability analysis showing multi-sequence prototype activations and shared prototype reuse across LGG/HGG cases (Appendix G, Page 29), demonstrating that similar tumour-related patterns can be captured across grades while their diagnostic effect is modulated by class-specific classifier weights.
>
> \textbf{Question 2:} We agree that extending evaluation to additional tasks, including multi-class classification, would further strengthen empirical validation. In this work, we focus on the established BraTS-2020 LGG vs HGG protocol to enable fair comparison with recent case-based baselines under the same setting. UM--ProtoShare is not task-specific, and the shared class-agnostic prototype formulation and multi-scale evidence aggregation can be naturally extended to multi-class and other medical imaging tasks; we highlight this as future work (Limitations and Future Work, Page 12).
>
> \textbf{Question 3:} To support the consistency of the CNN baseline results, we added Occlusion sensitivity as an additional experiment for post-hoc attribution method (Table 1, Page 10; Figure 2, Page 11; Comparative benchmark, Pages 10--11; Appendix F.2, Page 28).

---

> > ### Comment · Reviewer_JLd3 · 2026-01-31
> >
> > I thank the authors for their efforts and detailed answers. Most of my concerns are answered. I would like to maintain the current score.

---

### Official Review · Reviewer_3fXt · 2026-01-08

**Confidence:** 4
**Preliminary Rating:** 2
**Final Rating:** 3

**Summary:**

The authors study the problem of interpretable deep learning classification through the lens of case/prototype-based reasoning. They propose UM-ProtoShare, a novel method that leverages a bank of shared, class-agnostic, multi-scale prototypes. Their model is tested on BraTS 2020 and compared to several other prototype-based segmentation methods. It achieves superior performance compared to those baselines, both in terms of balanced accuracy, reaching accuracy close to classical (and "non-interpretable") CNN, and of Activation Precision and Incremental Deletion Score, which measure the interpretability.

**Strengths:**

- The paper contains many experiments, in particular to justify the many design choices of the proposed method (class-agnostic prototypes, fusion, multi-scale, etc.)
- The components of the model (prototypes, losses, similarities, etc.) are rigorously explained in the method section
- The supplementary pages provide insightful details for reproducibility. I encourage the authors to share their code if the paper is accepted, as they have stated in the article.
- The proposed model achieves better metrics than methods to which it is compared, both in terms of classification and interpretability.

**Weaknesses:**

- The methodological novelty is quite limited in my opinion as their proposed model is a light incremental improvement from previous similar methods (MProtoNet, MAProtoNet). The authors would benefit from stating more clearly the architecture improvements they introduced compared to those methods. The performance improvement is fairly limited, particularly given the large confidence intervals.
- The paper lacks clarity and is rather hard to follow. I acknowledge that the method is complex and involves many design components that need to be introduced but I believe it is complicated to follow the explanations of the architecture of the network and thus understand the importance of each component. I hope that the two additional pages allowed in the rebuttal phase will help improve the readability and clarity of the article.
- The Results part of the main body of the paper is short and not very insightful in my opinion. The major part of the discussion is left for the very last part of the supplementary material.
- The evaluation is limited to a single task, brain tumor classification. It would be beneficial to test it on other classification tasks (with ground truth segmentation masks for interpretability).
- The method contains several training stages and loss terms, and no details are given on how the weights of each loss term have been chosen. It is a potential problem for the generalizability of the method, and because the increase in performance compared to the baselines could be due to a more careful tuning of the hyperparameters.
- I feel like for a work that promotes interpretable methods and defends their value in a clinical setting, the analysis concerning interpretability is poor. It is limited to a purely quantitative analysis, with the incorporation of two metrics (IDS and AP) in the benchmark, and there is no qualitative interpretation of the prototypes, the assignements or the similarities between different cases.  One of the statements of the authors is to justify the interest of class-agnostic clustering over class-specific clustering is that "several MRI features (e.g., peritumoural oedema, necrotic cores, enhancing rims) may legitimately appear across tumour types and grades", but there is no experiment/visualization other than the improved metrics in the ablation that can convince a reader to believe that some of the prototypes are indeed commonly used by the two classes.

**Detailed Comments:**

- It is not clear how the total number of prototypes K has been chosen. It would be interesting to conduct an experiment that shows the influence of its value, as the parameter K could intuitively play a crucial role in the final performance.
- Table 1 gathers two types of information: the comparison with existing baselines and the ablation study. I would prefer to have the two separated for easier reading
- Figures 1 and could be easier to read if they were bigger.
- For the prototype-diversity regulariser loss, is there any justification for choosing to penalize the average pairwise cosine similarity? To my knowledge, constrative-based loss are also popular and could have been tested here, see [1] for instance

[1] T. Zhou, W. Wang, E. Konukoglu and L. Van Goo, "Rethinking Semantic Segmentation: A Prototype View," 2022 IEEE/CVF Conference on Computer Vision and Pattern Recognition (CVPR), New Orleans, LA, USA, 2022.

**Justification Of Final Rating:**

The authors have carefully responded to my questions and comments and have, I think, improved the quality and clarity of the paper. However, I still have a few reservations to recommend the acceptance of this work. First, I still believe that the methodological novelty is modest, although the appendix G supports the usefulness of shared prototypes. The proposed method outperforms on average the baselines but the confidence intervals are quite large. Figure 2 does not provide visual evidence of the potential benefits of their method in terms of interpretability as the attention maps are almost identical to those of MProtoNet and MAProtoNet, which could suggest the minor improvements in metric scores may not lead to significant interpretability improvement in a clinical setting. The evaluation is limited to a single dataset with large and rather easy-to-spot lesions. In particular, the proposed method relies on many hyperparameters in the model design and the training, and an evaluation on other datasets would be necessary to assess if the method is robust to other applications or if the parameters have been fine-tuned on BraTS to achieve superior metrics. For all these reasons, my final rating will be Borderline.

**Justification Of The Preliminary Rating:**

Even though the proposed method achieves better performance compared to several baselines, the methodological contribution is quite limited in my opinion, as the proposal of the authors seems to be a light increment and aggregation from previous prototype-based classification models (multi-scale approach, CAM-based module, prototype attention, etc.). The method is complex, it contains many components and loss terms, and it is validated on a single task, which limits its potential impact and generalizability. The interest of case-based reasoning for classification is motivated in the introduction, but the work lacks a proper qualitative evaluation of the use of prototypes.

**Questions To Address In The Rebuttal:**

The authors are encouraged to respond to the elements reported in the Weaknesses and Detailed Comments section.

---

> ### Author Response · Authors · 2026-01-24
>
> We thank the reviewer for their careful evaluation and constructive feedback. We have revised the manuscript accordingly, and provide point-by-point responses below (changes are highlighted in yellow in the revised manuscript).
>
> \textbf{Weakness 1:} We acknowledge that the proposed work builds on the literature in the case-based framework such as MProtoNet and MAProtoNet. This manuscript has clear contributions that are explored, analysed, and evaluated, rather than relying on incremental improvements that mainly depend on minor tweaks for performance gains. Unlike existing works, the proposed method presents a shared class-agnostic prototype bank, which is memory-efficient and scalable, class-independent, and naturally applicable to multi-class problems and other domains. Further, the proposed work uses a multi-scale encoder--decoder architecture, \textit{which has not been explored in this setting}. Gated fusion allows efficient feature integration while maintaining computational efficiency and improving localisation performance. The performance gains linked to these components are shown in Table 1 (Rows 7--14). For clearer contributions, we revised the Introduction (Page 3, final paragraph), and Figure 1 (Page 4) highlights UM--ProtoShare-specific components in blue.
>
> \textbf{Weakness 2:} The revised manuscript has been restructured for clarity, making the pipeline linear and self-contained in the main paper. We added an end-to-end roadmap (Model Architecture, Page 4) and revised Training objectives and loss functions to report the three training stages, stage-wise objectives, and all loss terms/weights (Section 2.3, Page 6). Standard class-specific formulations were moved to Appendix B.2 (Page 24), while the class-agnostic prototype-sharing formulation remains in the main text.
>
> \textbf{Weakness 3:} The Results section has been expanded to provide clearer takeaways in the main paper, including stronger interpretation of the ablation trends (Pages 9--10) and a clearer discussion of the accuracy--interpretability trade-off in the comparative benchmark (Pages 10--11). A concise Discussion section has also been added to consolidate the main insights (Pages 11--12).
>
> \textbf{Weakness 4:} We follow the established BraTS-2020 glioma grading protocol to enable fair comparison with prior case-based baselines (e.g., MProtoNet). We agree that extending evaluation to additional classification tasks (including mask-annotated datasets for stronger localisation assessment) would add clear value, and we therefore highlight this as future work in the Limitations and Future Work section (Page 12).
>
> \textbf{Weakness 5:} We acknowledge that the initial manuscript did not clearly report the loss-weight settings. In the revision, we list all loss coefficients and clarify that we adopt the same loss-weight settings as closely related baselines (e.g., MProtoNet/MAProtoNet), keep them fixed across folds, ablations, and the comparative benchmark, and introduce only UM--ProtoShare-specific terms, reducing the risk of hyperparameter tuning (Appendix D.5, Page 25).
>
> \textbf{Weakness 6:} We agree that AP/IDS would benefit from qualitative support for clinically oriented interpretability. We added a scale-wise qualitative analysis showing prototype behaviour across cases and evidence reuse across grades, including evidence patches, attention maps, and per-class logit contributions to illustrate shared class-agnostic prototypes and class-specific decision effects (Appendix G: Page 29; Figure 3: Page 31).
>
> \textbf{Detailed Comment 1:} We have revised the manuscript to clarify how K is chosen (Appendix A.3, Page 21). In this work, we fix K = 30 to ensure a capacity-matched and fair comparison across UM–ProtoShare ablations and baseline methods. We agree that a dedicated K sensitivity study would be valuable; in E1–E3, we already show how reallocating prototype budget across scales affects performance and interpretability. Due to scope and page constraints, we do not include a full sweep over K in this submission and consider it as future work.
>
> \textbf{Detailed Comment 2:} We agree this separation could improve readability; due to strict page limits we keep a single consolidated table, but moved Table 1 to improve the Results flow (Page 9).
>
> \textbf{Detailed Comment 3:} We have revised Figure 1 to improve readability by increasing its size (Page 4).
>
> \textbf{Detailed Comment 4:} In UM--ProtoShare, we adopt an average pairwise cosine-similarity penalty as a lightweight regulariser that discourages redundant prototypes in the same cosine-similarity space used for prototype matching (statement updated at the end of Page 8), consistent with prior prototype-based models (e.g., ProtoPool, ProtoLens, ProtoArgNet). We agree that contrastive-style objectives (Zhou et al.) are a strong alternative for promoting prototype diversity and separation, and we will consider contrastive-based diversity objectives as a promising direction for future work.

---

> > ### Comment · Reviewer_3fXt · 2026-02-01
> >
> > I would like to thank the authors for their answers; I have changed my rating to Borderline, the justifications are detailed in the official review.

---

### Official Review · Reviewer_KQq3 · 2026-01-15

**Confidence:** 3
**Preliminary Rating:** 4
**Final Rating:** 4

**Summary:**

This paper proposes UM–ProtoShare, an interpretable case-based model for pre-operative glioma grading (LGG vs HGG) from multi-sequence 3D MRI. It learns a shared, class-agnostic multi-scale prototype bank and predicts grades by matching image regions to these prototypes, producing “this-looks-like-that” explanations. Using a 3D ResNet backbone with UNet-style decoding, it is evaluated on BraTS-2020 with 5-fold cross-validation, showing performance close to a CNN baseline while improving interpretability metrics over prior prototype-based methods.

**Strengths:**

- The paper presents a novel prototype-based framework that improves interpretability while maintaining competitive balanced accuracy on BraTS-2020.
- The evaluations are thorough and solid, including comparisons to multiple baselines and component-wise ablations, and demonstrate the significance of the proposed model.

**Weaknesses:**

- Many interpretive details and discussions are deferred to the appendix. Adding at least a brief discussion section in the main paper (e.g., clinical implications, limitations, and future work) would make the contribution easier to assess. Right now, the lack of analysis or discussion in the main paper makes it seem not really self-contained.

**Detailed Comments:**

- Minor presentation suggestion: moving table 1 and figure 2 closer to their first discussion (sections 3.3 and 3.4) would improve readability; right now, they appear late and visually split the transition into the conclusion.

**Justification Of Final Rating:**

The authors have addressed all of my questions and concerns in the rebuttal, and their clarifications resolve the key issues I raised. I will keep my score unchanged. Overall, the manuscript is clearly written and reasonably justified.

**Justification Of The Preliminary Rating:**

The paper proposes a prototype-based framework that improves interpretability while maintaining competitive balanced accuracy on BraTS-2020 with extensive comparisons and ablations. My concern is that the main paper feels appendix-dependent, with many interpretive details and broader discussion pushed out of the main text, which reduces self-containedness. These are fixable issues, and I think the technical contribution is solid.

**Questions To Address In The Rebuttal:**

Please consider addressing the points in “weaknesses” and “detailed comments”.

---

> ### Author Response · Authors · 2026-01-24
>
> We thank the reviewer for their careful evaluation and constructive feedback. We have revised the manuscript accordingly, and provide point-by-point responses below (changes are highlighted in yellow in the revised manuscript).
>
> \textbf{Weakness:} We agree that deferring substantial interpretive details to the appendix can reduce the self-contained clarity of the main paper and make the clinical contribution harder to assess. In the revised manuscript, we strengthened the main narrative to ensure that the key takeaways are presented in the core text rather than being appendix-dependent. We expanded the Results discussion to provide clearer trend-based interpretation for both the Ablation studies (Pages 9--10) and the Comparative benchmark (Pages 10--11), and added a dedicated Discussion section summarising the main clinical implications of UM--ProtoShare (multi-scale, multi-sequence case-based evidence), together with limitations and future directions (Page 12). The appendix now serves as extended analysis rather than the primary location of interpretation.
>
> \textbf{Detailed Comment 1:} Table 1 (Page 9) and Figure 2 (Page 11) were moved to appear immediately after their first discussion in Sections 3.3--3.4 to improve readability and maintain a smoother narrative flow.

---

> > ### Comment · Reviewer_KQq3 · 2026-02-01
> >
> > I would like to thank the authors for the clarification and detailed explanation. I would like to maintain the current score.

---

### Author Rebuttal · Authors · 2026-01-24

**Rebuttal:**

We would like to thank the reviewers for their constructive and valuable feedback, and we greatly appreciate the support for our work. We have made every effort to incorporate as many suggestions as possible within the available revision time frame. We have uploaded a revised manuscript with changes clearly highlighted in yellow for ease of review.

**Supporting Material:**

/attachment/62703220d7a8242871001bfe703fc7d60e21c3df.pdf

---

> ### Author Response · Authors · 2026-01-26
>
> Dear Reviewers,
>
> Thank you again for your time and constructive feedback on our manuscript.
>
> I would like to note a minor LaTeX formatting issue in the revised manuscript.
>
> Due to the sizing/placement of Figure 1, the PDF extends by a few lines onto the beginning of page 13. This is purely a layout issue that can be fully resolved through formatting adjustments without changing any scientific content.
>
> Thank you so much for your understanding.

---

### Meta-Review · Area_Chair_MRrL · 2026-02-07

**Recommendation:** Accept (Poster)
**Confidence:** 2

**Metareview:**

Reviewers consistently appreciated the thorough experimental evaluation, including strong ablation studies, competitive balanced accuracy relative to non-interpretable CNN baselines, and the use of quantitative interpretability metrics (AP and IDS). The revised manuscript substantially improved clarity, self-containedness, and discussion of clinical implications, addressing earlier concerns about over-reliance on supplementary material.

The main reservations raised by reviewers concern the level of methodological novelty, which some view as incremental relative to prior prototype-based approaches, the reliance on a single dataset and binary classification task, and the limited qualitative interpretability analysis despite the paper’s clinical motivation. While these concerns remain partially valid, reviewers generally agreed that the authors provided reasonable justifications, improved the presentation, clarified loss design and training stages, and added qualitative evidence within the given constraints.

Given the predominantly weak accept to borderline evaluations, the constructive rebuttal, and the relevance to the MIDL community, I support acceptance as a poster presentation.

---

### Decision · Program_Chairs · 2026-02-13

Accept (Poster)